



**MUDA: dynamic geophysical and geochemical MUltiparametric DAtabase**

Marco Massa [1], Andrea Luca Rizzo [1,2], Davide Scafidi [3], Elisa Ferrari [1], Sara Lovati [1], Lucia Luzi [1] and MUDA WG (*)

[1] National Institute of Geophysics and Volcanology (INGV), Milano department, Italy

[2] University of Milano Bicocca, DISAT, Milano, Italy

[3] University of Genova, DISTAV, Genova, Italy

(*) A full list of authors appears at the end of the paper
Corresponding author: marco.massa@ingv.it
**Abstract**
In this paper, the new dynamic geophysical and geochemical MUltiparametric DAtabase
(MUDA) is presented. MUDA is a new infrastructure of the National Institute of Geophysics and
Volcanology (INGV), published on-line in December 2023, with the aim of archiving and
disseminating multiparametric data collected by multidisciplinary monitoring networks. MUDA is a
*MySQL* relational database with a web interface developed in *php,* aimed at investigating in quasi real
time possible correlations between seismic phenomena and variations in endogenous and
environmental parameters. At present, MUDA collects data from different types of sensors such as
hydrogeochemical probes for physical-chemical parameters in waters, meteorological stations,
detectors of air Radon concentration, diffusive flux of carbon dioxide ($CO_2$) and seismometers
belonging both to the National Seismic Network of INGV and to temporary networks installed in the
framework of multidisciplinary research projects. MUDA daily publishes data updated to the previous
day and offers the chance to view and download multiparametric time series selected for different
time periods. The resultant dataset provides broad perspectives in the framework of future high
frequency and continuous multiparametric monitorings as a starting point to identify possible seismic





precursors for short-term earthquake forecasting. MUDA is now quoted with the Digital Object
Identifier https://doi.org/10.13127/muda (Massa et al., 2023).
**Key words:** multiparametric data, monitoring networks, dynamic data base, earthquakes forecasting
**1 Introduction**
Today, the increasing awareness on the interaction between tectonics and crustal fluids
dynamics is still lacking a simultaneous monitoring of the relative key factors. Changes in water
chemistry and levels, spring discharges, soil flux regimes (e.g. $CO_2$, $CH_4$, Radon) and compositions
of dissolved gases in water are well-documented in the literature (e.g. Italiano et al., 2001, 2004;
Chiodini et al., 2020; Gori and Barberio, 2022 and references therein), as being pre-, co- and post-
seismic modifications as well as markers of the local tectonic stress acting in the crust. These
recognized seismic-induced variations in groundwaters and springs led, in recent years, scientists to
give more attention to the development of multiparametric monitoring, in order to capture the main
evidence concerning abrupt changes in chemical and physical parameters recorded before (and also
after) energetic seismic events (Rikitake and Hamada, 2003; Cicerone et al., 2009; Martinelli, 2018
and references therein). The ultimate goal is to find systematic signals that can be assumed as possible
"precursors" or indicators that a seismogenetic process is ongoing (Hubbert and Rubey, 1959; Brauer
et al., 2003; Miller et al., 2004; Chiarabba et al., 2009; Di Luccio et al., 2010; Malagnini et al., 2012;
Keranen and Weingarten, 2018; Napolitano et al., 2020; De Matteis et al., 2021; Gabrielli et al., 2022,
2023; Ventura and Di Giovambattista, 2013). At the Italian scale, several studies have described the
utility of groundwater and spring parameters and soil gas emissions to catch seismic-related signals
as well. However, only a few studies reported a continuous, high-frequency monitoring, mainly of
groundwater level or hydraulic pressure (De Gregorio et al., 2012; Barberio et al., 2017; De Luca et
al., 2018), or as Gori and Barberio (2022) concerning spring monitoring (i.e. temperature, pH,



electrical conductivity, dissolved oxygen and carbon dioxide) or D'Alessandro et al. (2020) on soil
radon emissions related to seismic activity.

MUDA (geophysical and geochemical MUltiparametric DAtabase), a new dynamic

multiparametric database published on-line in December 2023 at the web site https://muda.mi.ingv.it
(Figure 1), has been developed in such a framework. MUDA is a new infrastructure of the National
Institute of Geophysics and Volcanology (INGV, www.ingv.it) devoted to archive daily and distribute
quasi real time geophysical and geochemical multiparametric data recorded in continuous or near-
continuous mode at selected sites installed at the most tectonically active Italian areas (Figure 2).
MUDA was designed in the framework of INGV Dynamic Planet S2-project (i.e. 3D structure of Italy
from multidata analysis. Passive/active seismic, magnetic, magnetotelluric, electrical, gravimetric
prospecting, https://progetti.ingv.it/it/pian-din) and its development is ongoing in the framework of
the INGV Dynamic Planet GEMME project (Integrated Geological, gEophysical and geocheMical
approaches for 3D Modelling of complex seismic site Effects).

The need for an infrastructure capable of acquiring, storing, organising and publishing

multiparameter data in near real time arose following the installation of the Garda multiparameter
seismic network, PDnet (https://eida.ingv.it/it/networks/network/ZO), installed starting from 2021 as
part of the Task-S2 of the INGV Dynamic Planet project (Massa et al., 2021; Ferrari et al., 2024). In
this framework, MUDA collects information from different types of sensors, such as seismometers,
accelerometers, hydrogeochemical for physical-chemical parameters in waters, geochemical for
measuring the diffusive flux of carbon dioxide ($CO_2$) from the soil or detecting the air Radon
concentration, and meteorological stations. The aim is to constraint the influence due to exogenous
parameters in order to make potential correlations between seismic phenomena and variations
concerning monitored parameters (i.e. groundwater level, temperature, electrical conductivity, $CO_2$
soil flux, air Radon concentration; Barberio et al., 2017; Chiodini et al., 2020; Mastrorillo et al.,

2020).



The challenge of MUDA is to provide the end-user a high quality dynamic but also
simultaneous and continuous monitoring of groundwater physical parameters, meteorological data
and seismic signals, together with gas concentration such as Radon or soil $CO_2$-$CH_4$ fluxes (Figure
3). In order to furnish the main information for a detailed interpretation of local phenomena in the
framework of multihazard assessment, the multiparametric data are provided together with all
necessary stations and sites metadata, supplied with a complete geological and morphological
description (Figure 4).

## 83    2 Seismotectonic framework and seismicity


The multiparametric sites now included in MUDA are located in five main target areas (Figure
2): Lake Garda, eastern Alps, Po alluvial basin and Northern and Central Apennine chains (Table 1).
Concerning instrumental seismicity (http://terremoti.ingv.it/), in the last 40 years, thousands of small
to moderate energy seismic events (Figure 2) occurred in Northern Italy. Despite the low-to-medium
seismic hazard of the area (Stucchi et al., 2011), the high level of exposure (e.g. metropolitan areas,
industrial plants etc.), the local geological condition and the proximity of active buried seismogenic
structure (Burrato et al., 2012) make many portions of North Italy a medium to high seismic risk zone
(Massa et al., 2022b, Lai et al., 2020).
In particular, the Garda region (Area 1, Figure 2) is characterized by low-to-moderate
seismicity with the active tectonic regime located on the margin of the southern Alpine chain
controlled by the Africa-Europe convergence. The main active faults affecting the area consist of
mainly NNE-SSW trending thrusts (Galadini et al., 2001). For instance, the November 24, 2004,
Vobarno $M_w$ 4.8 earthquake (https://terremoti.ingv.it/event/1564989), generated maximum
macroseismic intensities (Imax) of VII/VIII (https://emidius.mi.ingv.it/CPTI15-DBMI15/, Locati et
al., 2022). It is worth noting that in the past the same area was struck by several powerful events, such





as the October 30[th], 1901 Salò $M_w$=5.4 earthquake (https://emidius.mi.ingv.it/CPTI15-DBMI15/,
Rovida et al., 2022).
Moving eastwards (Area 2, Figure 2), the highest rate of energetic events in Northern Italy is
associated to the South-verging thrust faults typical of the central and East South Alpine Chain
(Battaglia et al., 2004; Serpelloni et al., 2005; D'Agostino et al., 2008), due to the North-South
convergence between the Adriatic microplate and the Alps. The most recent destructive earthquake
occurred in Friuli, during the seismic sequence of May 6, 1976, with $M_w$= 6.5 (Pondrelli et al., 1999
and reference therein), whereas the largest historical event was the 1695 Asolo earthquake, with an
estimated $M_w$=6.48 (https://emidius.mi.ingv.it/CPTI15-DBMI15/) broadly associated to the thrust
system of the Montello area (Danesi et al., 2015).
South of the Alps, the Po alluvial plain (Area 3, Figure 2) represents a very deep foreland
basin of two opposing verging fold-and-thrust belts developing in the framework of the African and
European plates relative convergence (Pieri and Groppi, 1981; Bigi et al., 1990). Despite the flat
morphology, the Po plain is far from being an undeformed domain, since the outermost and most
recent thrust fronts of the two belts are buried by the Plio-Quaternary sedimentary sequence (Burrato
et al., 2012). The historical and instrumental Italian seismic catalogues show that the southern Po
plain is affected by low to moderate seismicity, with $M_w$ up to 5.8 during the 2012 sequence (Luzi et
al., 2013). Considering the historical seismicity (Rovida et al., 2020), the central part of the Po plain
was struck by the more significant North Italy earthquake on January 3, 1117, with an estimated
$M_w$=6.52.
Moving southwards, the Northern Apennines (Area 4, Figure 2) underwent the regional
seismicity associated with the Apennine fronts defined by different arcs of blind, North-verging
thrusts and folds (Mazzoli et al., 2015; Chiaraluce et al., 2017), capable of generating moderate
energetic seismic events with a maximum magnitude around 6 (i.e. June 5, 1501, $M_w$ 6.05, Rovida et
al., 2022). In particular, this area hosts the Nirano site (Table 1), in the Regional Natural Reserve of
Salse di Nirano (Giambastiani et al., 2024; Romano et al., 2023), an area lying upon an anticline





structure of the North-East verging fold-and-thrust Apennine belt characterized by one of the largest
mud volcano fields in Europe (Bonini, 2008; Castaldini et al., 2005) coupled to the emission of $CH_4$-
dominated gases (e.g., Buttitta et al., 2020).
Finally, two stations included in MUDA, are installed in the surroundings of the Norcia
alluvial basin (Area 5, Figure 2), an area characterized by high seismic hazard and seismicity rate due
to dense extensional NW-SE active fault systems (e.g. Galadini et al., 1999, Brozzetti and Lavecchia,
1994) capable of generating high-magnitude earthquakes (Galli et al., 2018; 2019), such as the
January 14, 1703, $M_w$=6.9, earthquake or other moderate events such as the 1328, $M_w$=6.3, the 1730,
$M_w$=5.9, the 1859, $M_w$=5.5 and the 1979, $M_w$=5.8, earthquakes (Rovida et al, 2022). The recent
instrumental seismicity highlights the two main events occurring on August 24, 2016 and on the
October 30, 2016, with $M_w$ 6.0 and 6.5 respectively, in an area a few kilometres of the Norcia plain
(e.g. Improta et al., 2019 and reference therein).

**3 State of the art**

At present, in Italy and Europe, the seismological communities in general are fairly advanced
in their running of both network data management and seismic data sharing. In Italy, the main seismic
network is represented by the National Seismic Network (RSN,
https://eida.ingv.it/it/networks/network/IV, Margheriti et al., 2020), managed by INGV and
sometimes integrated for real time data exchange by many local or regional networks (Massa et al.,
2022). The RSN permanent network is codified through the IV code assigned by the International
Federation of Digital Seismograph Networks, FDSN (https://www.fdsn.org/). The RSN station codes
are registered at International Seismological Center, ISC (http://www.isc.ac.uk/), while data, recorded
following the SEED (Standard for the Exchange of Earthquake Data,
http://www.fdsn.org/seed_manual/SEEDManual_V2.4.pdf) format, are shared (Danacek et al., 2022)
through the EIDA-Italia node (European Integrated Data Archive, https://eida.ingv.it/it/). In Italy,





INGV provides many websites and thematic databases for real time data quality and distribution,
such as EIDA-Italia, ISMDq (INGV Strong Motion Data quality, https://ismd.mi.ingv.it , Massa et
al., 2022), ITACA (ITalian Accelerometric Archive, https://itaca.mi.ingv.it, Pacor et al., 2011), ESM
(Engineering Strong Motion database, https://esm-db.eu/, Luzi et al., 2016), BSI (Italian Seismic
Bulletin, https://terremoti.ingv.it/bsi, Marchetti et al., 2016), TDMT (Time Domain Moment Tensor,
https://terremoti.ingv.it/tdmt, Scognamiglio et al., 2009), ShakeMaps (https://shakemap.ingv.it/,
Michelini et al., 2020), etc.
Differently, the geochemical community has still not developed a so capillary network of automatic
stations for data acquisition, management and sharing, as the seismic community does. This mostly
depends on the fact that only a few geochemical parameters/tracers can be measured directly in the
field and in near real-time [e.g., diffusive flux of $CO_2$ from the soil through accumulation chamber
method (Chiodini et al., 1998; Carapezza et al., 2004; Inguaggiato et al., 2011a; Rizzo et al., 2015),
radon concentration in atmosphere or from the soil with specific Geiger counters, concentration of
$H_2O$, $CO_2$, $SO_2$, $H_2S$, $CH_4$, halogens in atmosphere through MULTIGAS sensors (Aiuppa et al., 2005;
Shinohara et al., 2005) or FTIR technique (e.g., Allard et al., 2005), $SO_2$ flux in atmosphere by DOAS
and UV techniques (Burton et al., 2009; Aiuppa et al., 2021)]. It must be also highlighted that most
of the automatic measurements of geochemical parameters above reported were developed and
applied in volcano monitoring, while only recently the geochemical community is moving to apply
some of those tracers to seismic monitoring. In terms of hydrogeochemical monitoring, apart the
physical-chemical parameters in water (e.g., temperature, water level, electric conductivity, and
others such as pH and Eh but with less precision and accuracy) for which automatic sensors exist
since long time, the automatic and high-frequency measurement of the water's composition is limited
to a few sensors developed in the last decade or so, which mostly focus on the concentration of a few
gas species dissolved in waters (e.g., $CO_2$, $CH_4$, total gas pressure; De Gregorio et al., 2005,
Inguaggiato et al., 2011b). As for gas sensors, most of the automatic measurements of water's
composition were developed for volcano monitoring applications.



At present, in Italy, hydrogeochemical and geochemical seismic monitoring is limited to
selected areas or sites, and it is essentially performed by several departments of INGV in the
framework of individual initiative such as the Alto Tiberina Near Fault Observatory (TABOO,
https://ingv.it/en/monitoring-and-infrastructure-a/monitoring-networks/the-ingv-and-its-
networks/taboo, Chiaraluce et al., 2014) or recent and on-going INGV projects such as the Dynamic
Planet (https://progetti.ingv.it/it/pian-din), FURTHER (https://progetti.ingv.it/en/further), MYBURP
(https://progetti.ingv.it/it/pian-din#myburp-modulation-of-hydrology-on-stress-buildup-on-the-
irpinia-fault), Multiparametric Networks or Rebuilding Central Italy, DL50 and  related tasks (e.g.
Idro-DEEP $CO_2$, Idro Calabria, Idro Nord), concerning the groundwater continuous monitoring (e.g.
springs and thermal waters) of different areas of mainly Central and Southern Italian Apennines
(https://www.pa.ingv.it/index.php/progetti/)    and    Radon    monitoring    (IRON    project,
https://ingv.it/monitoraggio-e-infrastrutture/reti-di-monitoraggio/l-ingv-e-le-sue-reti/iron).    In
particular, the Alto Tiberina Near Fault Observatory is managed by EPOS (European Plate Observing
System) research infrastructure (https://www.epos-eu.org/) of which the mission is to foster the
integration of solid earth data and their by-products made by the entire European scientific
community: in this case, seismological, geophysical, geodetic and geochemical data recorded by
TABOO-Near Fault Observatory are accessible via the FRIDGE European web portal
(https://fridge.ingv.it/index.php).
Further local monitoring initiatives are provided by other institutions or Universities through
the installation of geochemical stations and probes in different parts of the national territory such as
in Tuscany, by the IGG-CNR (Institute of Geoscience and Earth Resources,
https://www.igg.cnr.it/en), in Southern Italy, by the IMAA CNR (Institute of methodologies for
environmental    analysis,    https://www.cnr.it/en/institute/055/institute-of-methodologies-for-
environmental-analysis-imaa) or Central Italy, by the Earth Sciences Department (DES,
https://www.dst.uniroma1.it/en) of Sapienza University of Rome (Martinelli et al., 2021).





Consequently, at a national scale the hydrogeochemical and geochemical monitoring is not
organized by using an ad hoc reference institutional database or web portal able to homogeneously
archive and distribute high quality multiparametric data to the scientific community. At present and
at the best of our knowledge, the only existent databases focus only on mapping gas emissions (e.g.,
MaGa, http://www.magadb.net/) or thermal springs, without archiving data from a regular
monitoring. A first attempt was recently made in the framework of an agreement between the National
Institute of Geophysics and Volcanology (INGV) and the National System for Environmental
Protection (SNPA, https://www.snpambiente.it/, comprising the Regional Environmental Protection
Agencies, ARPA, and the Italian Institute for Environmental Protection and Research, ISPRA,
https://www.isprambiente.gov.it/), with the aim of sharing data from the continuous monitoring of
water wells and springs,  in particular the piezometric level, temperature, electrical conductivity,
salinity and total dissolved solids (Comerci et al., 2019).

**4 MUDA database**

MUDA is a dynamic and relational multiparametric database designed and built using a table
structure that can correlate data of a different nature (i.e. seismic, hydrogeochemical, geochemical,
meteorological). It is adaptable to further types of data from other projects and capable of integrating
perfectly with those already acquired via both real time and off-line transmission vectors (Figure 5).
The MUDA database is based on MySQL (https://www.mysql.com/it/), a popular and efficient open
source relational database management system for handling large amounts of data. Particular attention
has been paid to optimising and, above all, integrating all the different types of data taken from
different sources while trying to maintain a certain structural uniformity, also open to possible future
new implementation.
Data collection takes place separately for each type of monitoring station (Table 1), each according
to its preferred channels (email, ftp system, Application Programming Interface API, Structured
Query Language SQL) in a effort to improve each procedure, avoid data loss and minimise the time
taken to receive data.
Data are acquired and then archived on a centralised server from which all pre-processing procedures
are then carried out to insert this data, after appropriate checks and automatic analysis, into the MUDA
database (see next chapter for details). All data downloaded from the remote stations, after the check
and processing phase are stored in files before the population of the MySQL database. This is also
convenient to have a native and complete data backup, for future requirements.
The MUDA database is structured to consider all the different types of monitoring stations at the same
multiparametric site through a univocal internal site code, linked to all different types of data. At the
same time, however, the independence of the data of each different station is also maintained, as each
individual site may have its own particular condition and metadata. For each type of monitoring
station, the MUDA database includes 2 tables, one for the station metadata and the other for the
recorded data, linked by a unique station id.

**4.1 Processing of raw data**

The MUDA project currently includes 5 types of data: hydrogeochemical, meteorological,

Radon, $CO_2$, and seismic. All data are pre-processed to align each time series to the common UTC
(Coordinated Universal Time) time. Hydrogeochemical, seismic, meteorological and Radon data are
moreover resampled in order to have a representative data each few minutes (i.e. from 1 to 5), namely
a good enough interval to see possible cross-correlation signals on different parameters. A data
resample is a priori necessary in order to at first homogenize data for viewing and comparison, but
also to allow the web page to have a fast response to any query involving long time periods (actually
up to a maximum of 30 days) of continuous and high-frequency multiparametric recordings. In
particular, while hydrogeochemical, gas and meteorological data are uploaded into MUDA database
as raw data with only a consistency check, the seismic data are pre-processed in order to obtain





waveforms metadata to be included into MUDA database and to be easily comparable in terms of
time series to the other parameters.
The processing of raw data for each of the 5 parameters included in MUDA is described in detail in
the following.

The hydrogeochemical data are acquired by two different type of instrumentation, the first

provided by Van Essen Instruments (https://www.vanessen.com/) and the second by STS-Italia S.r.l.
(https://www.sts-italia.it/). Data recorded by Van Essen Instruments and STS-Italia are set to sample
records every one and ten minutes, respectively. In both cases, groundwater level (m), electrical
conductivity (µS/cm), and temperature (C°) are achieved using probes (e.g. CTD-Diver®,
https://www.vanessen.com/) installed in correspondence of wells or springs by using weir
flowmeters; the recorded chemical and physical parameters are sent at defined time intervals to the
remote company head offices and then to the INGV acquisition centre by proprietary API or email.
At present, remote stations send data in ASCII format twice a day (i.e. 9 a.m. and 9 p.m., Central
European Time, CET) as an email attachment or by ftp-protocol. Depending on the target
instrumentation, before populating the MUDA database, a pre-processing is necessary: as an example,
the Van Essen probes for water wells need a barometric compensation to account for atmospheric
pressure variations in order to provide the corrected water level value (Ferrari et al., 2024). Automatic
data acquiring and processing tools have been developed in Python (https://www.python.org/), while
automatic tools for populating the MUDA database have been developed in PHP language
(https://www.php.net).

The meteorological data included in MUDA are provided by Davis Vantage Vue®

instrumentation (https://www.davisinstruments.com/). Each single meteorological station, placed
near the water well, provides information on atmospheric pressure (mbar), temperature (C°), humidity
(%), rainfall amount (mm), wind speed and direction. Also in this case, data take advantage of
GPRS/LTE technologies and are gathered by a dedicated WeatherLink Live cloud platform, making
them available in real time on the dedicated website (https://www.weatherlink.com/). With samples



every one minute, data are archived into the Davis server-cloud and then shared by payment to the
end-users (e.g. INGV server) through proprietary Application Programming Interface (API) set into
an automated ad-hoc developed python tool, suited for all the different kinds of meteorological
instrumentation at each site. Just as for hydrogeochemical data, meteorological parameters are then
automatically inserted into the MUDA database twice a day (i.e. 9 a.m. and 9 p.m. CET) using a
procedure developed in PHP.
The Radon data are provided by the IRON network (Italian Radon mOnitoring Network,
https://ingv.it/monitoraggio-e-infrastrutture/reti-di-monitoraggio/l-ingv-e-le-sue-reti/iron, Cannelli
et al., 2018). Stations, placed next to the water well, measure the concentration of gas in the air using
a photodiode detector (AER Plus, Algade), with a sensitivity of 15 Bq/m$^3$ by counts/hour. Data are
measured and acquired every 4 hours together with temperature and humidity, Radon data are
transmitted in real time by the Sigfox (https://www.sigfox.com) 0G-technology and archived at the
Sigfox cloud, with the exception of particularly remote sites where a periodic local data downloading
is also necessary. Radon data are provided in csv format, where the concentration is measured in
Bq/m$^3$, and then uploaded into the MUDA database using an ad-hoc developed PHP tool. In this case,
the procedure is manually started after each single data downloading.
The $CO_2$ soil flux measurements are acquired using an accumulation chamber provided by
Thearen S.r.l (https://thearen.com/). The permanent stations have a no-stationary flux chamber and
are equipped with an infrared analyser measuring $CO_2$ concentrations ($g \cdot m^{-2} \cdot d^{-1}$) in a time frame of
three minutes. A single $CO_2$ flux measure is returned each hour already corrected for pressure (mbar)
and temperature (C°) recorded inside the chamber. Soil temperature and humidity (%) and
meteorological parameters (atmospheric pressure, temperature, humidity, rain, wind speed and
direction) are acquired concurrently. Data are sent to the head company server-cloud through a
dedicated modem with automatic data transmission. Data are acquired daily by the INGV acquisition
centre through an ad-hoc server to server link using an internal INGV-VPN (Virtual Private Network)



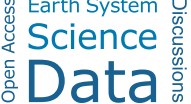

connection. Data provided in csv format, are then daily automatically inserted into the MUDA
database by using an ad-hoc developed PHP tool.

The Seismic data are acquired by selected stations of the Italian National Seismic Network

(RSN, https://eida.ingv.it/it/networks/network/IV) and the multiparametric network of Northern Italy
(PDnet,   https://eida.ingv.it/it/networks/network/ZO)   placed   near   the   water   well   used   for
hydrogeochemical data. Recorded data are codified following the international standard commonly
used by the seismological community, namely the FDSN (https://www.fdsn.org/) network-station
code       and       SEED       (Standard       for       the       Exchange       of       Earthquake       Data,
http://www.fdsn.org/seed_manual/SEEDManual_V2.4.pdf)   format   supported   by   European
Integrated Data Archive, EIDA (https://eida.ingv.it/it/) and maintained by the FDSN. Data are
transmitted by different technologies (LTE, satellite, etc.) to the INGV Milano acquisition centre
where they are archived through a Seiscomp4 (https://www.seiscomp.de/doc/apps/seedlink.html)
client to improve the SeedLink real time data acquisition protocol. Data are archived in the standard
binary miniSEED format (http://ds.iris.edu/ds/nodes/dmc/data/formats/miniseed) and organized in a
structured archive. Seismic data are pre-processed every night considering the 24 hours of all
miniSEED files recorded by stations on the previous day and then checked for quality before being
automatically included in the MUDA database by using an ad-hoc developed PHP tool.

**4.2 Seismic data processing**

Seismic data collected by MUDA come from permanent and temporary networks able to send

data in real time to the INGV acquisition centres. Following the international standard shared among
the seismological community, continuous data streams are usually archived in miniSEED format.
Depending on the adopted sampling rate, usually 100 Hz for seismometers and 200 Hz for
accelerometers, the amount of data per day relative to a single component data stream of motion at
one station ranges between about 6 and 10 Mbyte/day for seismometers and between 15 and 20

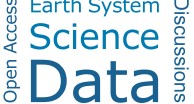

Mbyte/day for accelerometers. Multiplying this amount of data by a large number of stations, the
result is a daily amount of data that easily increases in the order of Gbytes and that is not easy to
manage in the framework of an immediate distribution provided by a thematic web portal.
Considering the main goal of MUDA, namely a daily dynamic updating web portal, for comparing
and downloading all multiparametric real time continuous data, and also considering the sampling
rate of the other available instrumentation (i.e. spanning from one record per minute to one record
every four hours), the continuous seismic data streams are processed in order to conform the contents
to the web portal requirements before inclusion in the MUDA database.
Seismic data are acquired and stored following the standard Seiscomp archive structure. Every night
at 2 a.m. miniSEED files relative to the 24-hour recordings of the previous day are selected at each
station and processed by using an ad-hoc procedure developed by merging the Bash and SAC
(Seismic Analysis Code, https://ds.iris.edu/files/sac-manual/) scripting languages.
The processing scheme starts by downloading at each single recording station 24 hours of miniSEED
files; data are then separated into 288 sub-windows, each one with length of 5 minutes starting from
the origin time of each single miniSEED file (usually corresponding to the 00:00:00 UTC if the station
works well). Then, for each 5-minute windows, raw data recorded in counts are converted to the
proper unit of measurements (cm/s for seismometers and gal for accelerometers) and the sensor
response curves are removed by deconvolution and finally filtered using a 4$^{th}$ order Butterworth filter
in the range 0.1 - 20 Hz. For each single sub-window relative to a specific channel recorded by a
specific station, the RSM (Root Mean Square, e.g. Goldstain et al., 2003), the maximum ground
shaking in terms of velocity (cm/s), the mean amplitude value of the whole FFT (cm/s/Hz) (Fast
Fourier Transform, e.g. Bormann 2012) and the maximum amplitude of the FFT for frequency
interval spanning from 0.1 Hz to 20 Hz, are calculated, providing for each parameter 288 values per
day, corresponding to the daily upload of  hydrogeochemical and meteorological data. In particular,
the Root Mean Square is calculated for the entire time windows using the following equation:



$$x_{RMS} = \sqrt{[1/n\,(x1^2 + x2^2 + ... + xn^2\,)]} \qquad (1)$$

where $x$ is the amplitude of the single sample, and $n$ the number of samples of the trace considered.
Daily time series of RMS, PGV, FFT-mean and FFT$(f)$ are finally uploaded into the MUDA database
using an ad-hoc developed PHP tool.
**4.3 Data availability and dissemination**

MUDA publishes and shares the available data recorded at each site through a specific web
interface developed in PHP (https://www.php.net/) to easily and effectively interact with MUDA SQL
database, and using a responsive design in HTML5, capable of adapting automatically to any device
on which it is displayed (i.e. PC, tablet, smartphone, etc.). As a final step, the data publication required
assigning a regular DOI associated to the DB and provided by INGV data management office through
a standard procedure. The final DOI of MUDA is https://doi.org/10.13127/SD/ku7Xm12Yy9. Data
have been licensed using the Creative Commons License CC BY 4.0. MUDA web portal publishes
multiparameter data daily and updated to the previous day. It offers the chance to view and download
dynamic time series for all available data and for different time periods, up to a maximum of 30 days.
In reference to longer periods, an e-mail request can be sent to: muda@ingv.it.
The web portal has a main page showing an interactive map of Northern and Central Italy, as
at present MUDA acquires data from automatic stations located in this part of the Country, where the
multiparametric stations are indicated by triangles with pop-ups showing the main features (i.e.
coordinates and available instrumentation) of the target site, including the direct access to the dynamic
data viewing (Figure 3) and to the single station page (Figure 4). The home page on the top right
corner shows a pop-up menu with selectable thematic layers including the reference seismic hazard
map of the national territory in terms of peak ground acceleration (MPS04 working group,
http://zonesismiche.mi.ingv.it/, Stucchi et al., 2011), the seismogenic areas and the active faults taken



from the Database of Individual Seismogenic Sources database (DISS, https://diss.ingv.it/, DISS
working    group    2021),    recent    (https://terremoti.ingv.it/)    and    historical
(https://emidius.mi.ingv.it/CPTI15-DBMI15/) seismicity bulletins, and the location of seismic
stations of the National Seismic Network (RSN, https://eida.ingv.it/it/networks/network/IV) managed
by INGV.

The *View & Download DATA* web page, accessible from the horizontal tool bar of the home

page, opens the dynamic data viewing (Figure 3). Users can select for each multiparameter site the
time series to be displayed backwards in time (i.e. 1 day, 7 days, 15 days, 30 days) starting from a
selected date. Once the site and period have been chosen, the available data automatically appear
synchronised with respect to UTC (Coordinated Universal Time). From top to bottom, these are:
hydrogeochemical data from well or spring sensors showing water temperature (°C), electrical
conductivity (µS/cm) and the value of the water column (m) above the sensors; meteorological data
showing air temperature (°C) and rainfall (mm); seismometric data showing the Root Mean Square
(RMS), the maximum ground velocity values (cm/s), the average Fourier spectrum (FFT) amplitude
values, the Fourier spectrum values for frequency bands selected in the interval 0.1 - 20 Hz; Radon
gas emissions ($Bq/m^3$) and soil $CO_2$ flux ($g{\cdot}m^{-2}{\cdot}d^{-1}$). All interactive graphs can be zoomed with the
left mouse button and they enable selecting individual functions using pop-up layers. In each graph,
in the top right-hand corner, it is possible to view the individual image in full screen and download
the selected data in csv (Comma Separated Values) format as well as the images in pdf (Portable
Document Format), png (Portable Network Graphic), jpeg (Joint Photographic Experts Group), svg
(Scalable Vector Graphics) formats. The last selectable item, on the right of this page, gives the
possibility of viewing a single parameter, for a more detailed observation.

A further topic of the MUDA web portal is the single station web page (i.e. link *STATIONS*),

also reachable from the horizontal tool bar. This web page is designed to provide a first and general
geophysical and geochemical characterization of each multiparametric site. In particular, each single
station web page shows, on the left, a thematic map indicating the location of the monitoring site.



Below the map, the two links provide a geological and morphological setting of the area. For each
recording site, a portion selected from the geological map at a 1:100,000 scale (Società Geologica
Italiana, http://www.isprambiente.gov.it/it/cartografia/) is provided, with topographic base at
1:25,000           scale           (Istituto           Geografico           Militare,
http://www.igmi.org/prodotti/cartografia/carte_topografiche/). Concerning morphology, for each
recording site, a topographic map (i.e. Slope and Ridge) is proposed by considering the available
digital   elevation   model   (ASTER   GDEM   with   a   cell   size   of   10   m,
https://www.earthdata.nasa.gov/news/new-aster-gdem). Starting from the processed DEM, the slope
map was constructed with three topographic classes (0°–15°, 15°-30°, and >30°), considering the
break values defined in the current Italian seismic code (NTC, 2018). The ridgelines were extracted
using the Topographic Position Index (TPI) algorithm (Weiss, 2001; Pessina and Fiorini, 2014). On
the right, the web page shows thematic tables relative to the installed instrumentation. At each site,
besides the general information on coordinates and technical features of the instruments, a
geophysical characterization of the site is also provided in terms of polarized horizontal to vertical
spectral ratio (HVSR) and each single log performed in the wells regarding water electrical
conductivity and temperature as a function of the available stratigraphy together with the main
features of monitored well or spring. Finally, in each station web page, graphs relative to the whole
hydrogeochemical time series are also downloadable.

**5 Data Records**

At present (i.e. 31 March, 2024), MUDA includes data from 25 multiparametric sites (Table

1) located in Northern and Central Italy (Figure 2), both already monitored by permanent INGV
infrastructures or installed in the framework of recent INGV research projects. It is worth mentioning
that not all multiparametric sites are characterized by homogeneous multiparametric instrumentation
(Figure 6). In any case, all available data are always sent in real time to the INGV acquisition centres.



In particular, concerning seismic data, the multiparametric sites include 7 stations belonging to the
permanent National Seismic Network (RSN, https://eida.ingv.it/it/networks/network/IV) and 12
stations belonging to the temporary multiparametric network of Northern Italy (PDnet, FDSN code
ZO, https://eida.ingv.it/it/networks/network/ZO), installed in the framework of the INGV Dynamic
Planet S2-project. These sites are located in Northern Italy, around the Lake Garda, at the southern
limit of the Eastern Italian Alps and in the central portion of the Po Plain (Figure 2). A further 2
seismic stations (PDN11, PDN12, table 1) have recently been installed in the framework of the INGV
Dynamic Planet GEMME (Integrated Geological, gEophysical and geocheMical approaches for 3D
Modelling of complex seismic site Effects) project, in Norcia basin (Apennine chain in central Italy)
and its surroundings (Figure 2). Finally, one seismic station (PDN10, table 1) has been installed in
cooperation with the Dynamic Planet PROMUD (Definition of a multidisciplinary monitoring
PROtocol for MUD volcanoes) project, in the Salse di Nirano Reserve (Italian Northern Apennines,
Figure 2).
In general, seismic stations are equipped with high dynamic 24-bit digital recording systems
coupled to enlarged (5s owner period) or broadband (120s owner period) seismometers. In particular,
at Oppeano multiparametric site (Table 1), borehole instrumentation is installed at 150 m depth. Even
if in some cases accelerometer sensors are coupled to the target seismmometer (velocimeter), MUDA
includes only seismmometric data due to their higher sensitivity and best resolution with respect to
possible occurrences of natural phenomena such as micro seismicity, local weak motion earthquake
occurrences and/or teleseisms or environmental modifications (i.e. landslides, low atmospheric
pressure, tides etc.).
Hydrogeochemical data are recorded in all 25 multiparametric sites (Table 1), at both wells
and springs set up in the framework of INGV Dynamic Planet projects (S2 and GEMME) and INGV
Multiparametric Networks or the Rebuilt central Italy, DL50 project. The Toppo site (Table 1),
installed in the eastern Alps in the framework of the Eccsel Eric consortium
(https://www.eccsel.org/), is managed by OGS (National Institute of Oceanography and Applied





geophysics, https://www.ogs.it/en) in cooperation with the University of Ferrara (Department of
Physics and Earth Science, http://fst.unife.it/en) and the INGV. At present, data included in MUDA
are recorded at 22 wells and 3 springs (Recoaro, Recoaro1, Feltre, see table 2). In general, wells have
depths ranging from a few meters (< 10 m) to a maximum of 300 m for Toppo and Mirandola sites
(Table 2). The instrumented wells have a mean depth between 5 and 150 m. In particular, the
monitored well named TRIPONZO (Table 2) is characterized by the presence of thermal waters.
All sites monitored by hydrogeochemical instrumentation are also equipped with a
meteorological station able to capture the main atmospheric variations.
Four sites (i.e. Montelungo, Bondo, Norcia, Triponzo, table 1) are currently equipped with
instrumentation able to record the Radon concentration in the air. Sensors for Radon monitoring are
part of the IRON network (Italian Radon mOnitoring Network, https://ingv.it/monitoraggio-e-
infrastrutture/reti-di-monitoraggio/l-ingv-e-le-sue-reti/iron) of INGV. Finally, just one site (i.e.
Nirano, table 1) is equipped with instrumentation for Carbon Dioxide ($CO_2$) soil flux measurements.
In general, the multiparametric sites show co-located instrumentation, with a few exceptions
due to logistic difficulties during the site installation or other technical problems (e.g. sites with a
very high level of background seismic ambient noise or working pump installed in well). In all cases,
the reference seismic station is installed in the same geological, morphological and hydrogeological
context as the other instruments, possibly co-located or at least in the proximity. Hydrogeochemical
stations installed in a narrow area (e.g., Bulgarelli, Medolla, Mirandola and Secchia sites, table 2)
might refer to the same seismic station.

**5.1 Data quality check**

In order to verify the completeness and correctness of the recorded data, we carried out several
checks.





Concerning seismic data included in MUDA, the results of the processing procedure to
produce 5 minutes interval data are verified to check the capacity of the proposed processing scheme
to represent a real marker to detect earthquakes or environmental phenomena. Figure 7 reports an
exhaustive example at the Oppeano site for events recorded by the seismic station on December, 29,
2020. Figure 7 shows in the panel $a$ (top) the occurrences of subsequent events in a narrow time
window spanning from 2020-12-29 14:02:40 to 15:36:57 UTC. The first evident transient is related
to the regional $M_w$=6.3 Croatia earthquake on 2020-12-29 at 12:19:54 UTC
(https://terremoti.ingv.it/event/25870121). A few hours later, a small sequence of three local weak
motion was recorded 12 km South West of Oppeano and localised by the INGV bulletin at 2020/12/19
14:02:40, 14:44:51 and 15:36:47 UTC with magnitudes of $M_L$=3.4, $M_L$=2.8 and $M_w$=3.9, respectively
(https://terremoti.ingv.it/event/25871441). In the bottom of panel $a$ the results of our detector
procedure in terms of FFT interval is shown for the 288 data points of the 2020/12/29. It is evident
how all transients have been clearly recognized by marked peaks: in particular, the regional event,
occurring about 400 km East of the Oppeano well highlights a notable contribution at low frequency,
showing a clear peak for the FFT around 0.1 Hz. On the contrary, the local seismicity is well described
by peaks detected at higher frequency content, in particular ranging from 1 to 10 Hz. In this case, it
is worth noting how the high frequency content of FFTs also highlight variations in ambient noise
level between night and day. In the panel $b$ of figure 7, the results of the detection procedure are
presented in terms of relative RSM, ground motion velocity and averaged FFTs (from top to bottom).
In order to publish reliable results, all continuous seismic data streams for all seismic stations
are checked daily for quality by the interoperability between MUDA and ISMDq (INGV Strong
Motion Data quality, http://ismd.mi.ingv.it, Massa et al., 2022). In this way all seismic stations
included in MUDA are checked for gaps (%), availability (%) and ambient noise level variation in
terms of PSD (Power Spectral Density) and PDF (Probability Density Function), in dB, as calculated
by McNamara and Buland (2004). Through ISMDq, it is possible to build temporal time series with
a maximum time length of up to 2 years for all stations included in MUDA in order to check at first



the correct functioning of the stations, the accuracy of used metadata, daily and seasonal variations
of ambient noise level and transient or permanent anthropic disturbances. In particular, in case of
failure in data transmission, the continuous monitoring of data gaps and availability allow us to
retrieve data directly from station memories thereby avoiding important gaps in the data.

Hydrogeochemical and geochemical data are checked daily for availability and gaps usually

due to a temporary lack of data transmission platforms (i.e. cellular line or satellite), in particular
during rainy and stormy days or more rarely due to malfunctioning of the instrumentation. In the first
case, the recorders, thanks to their internal memory and datalogging capacity, are able to archive data
up to a maximum of 30 days. If needed, the recovery of data is possible through a manual download.
Subsequently data are uploaded into the MUDA database by calling the same ad-hoc developed tools
for populating the database for each parameter, with appropriate flags. In some cases, water level time
series show unusual abrupt peaks (spikes) due to some problem during the compensation of
atmospheric pressure performed by the recording system in order to obtain the correct values of water
level: in general, misalignments in pressure compensation lead to wrong water level values, usually
with differences of ~ 10 m, as a consequence of the measured atmospheric pressure usually around
or slightly above 1000 mbar, considering that all the stations are within a few hundred meters above
sea level.

Other data included in MUDA (i.e. meteorological, Radon, $CO_2$ flux) do not need particular

processing. In any case, for all meteorological, Radon and $CO_2$ measurements, data are always
checked for gaps and possible spurious peaks that should be deleted. In particular, meteorological
data regarding the pluviometry are archived and uploaded into MUDA as a single sample recorded
each minute, or even better, in some cases, at each individual movement of the rain gauge's tipping
bucket (0.2 mm every time). In these cases, before publishing the rain values on the web site, data are
cumulated in intervals of 1 hour directly in the SQL query made by the web portal, in order to better
highlight heavy storms or other particular meteorological phenomena.





## 6 Multiparametric monitoring

**6 Multiparametric monitoring**

In this session, some examples of comparison among multiparametric data are presented and
discussed in the framework of their possible applications for research and services devoted to natural
hazard risk reduction.

An interesting example of multiparametric monitoring, concerning groundwater level

variation presumably related to a large landslide phenomena has been collected at the Bondo (Table
1) site and aquifer (Lake Garda area, Figure 2), where on the November 1, 2023, the water column
above the sensor in the well abruptly increased by ~ 20 m. This significant modification, also
combined with the diminishing temperature and electrical conductivity (Figure 8, panel *a*), happened
as a consequence of 2 days of intense rainfall with measured values of precipitation in a narrow area
surrounding Bondo up to 400 mm. It is important to highlight that the stratigraphy below Bondo is
mostly made of fractured-carsified limestones belonging to the Dolomia Principale formation (Upper
Triassic). To explain such a notable groundwater level variation, additional and contemporary natural
phenomena should be hypothesized; on the basis of the information provided by the local media, in
the same period the area was affected by diffuse landslides. In particular, between October 31 and
November 1, 2023, a clear seismic transient (Figure 8, panel *b*) was recorded on all 3 components of
ZO.PDN3 station (Table 2), characterized by very long duration (i.e. some hours), higher amplitudes
with respect to the local ambient noise level (i.e. the Peak Ground Velocity, PGV was equal to 0.01
cm/s, comparable to a local earthquake with magnitude ranging from 2.5 and 3.0), prevalent high
frequency content (~ 5-30 Hz) and a strong polarisation with preferential amplification of motion
along the NS direction (Figure 8, panel *c*). In this case, the recorded transient at ZO.PDN3 could in
fact be attributable to local and diffuse landslides that could have modified the volume and/or the
extent of the aquifer and eventually the water flow infiltration and circulation through the rock
fractures by influencing the water level and leading the aquifer to be more sensitive to meteorological
events.





Meteorological events, in particular the intense rainfall period, also seem to have a strong
influence on Radon emission measurements, usually adopted as a possible marker in case of a tectonic
event. An example can be observed considering the complete Radon time series at the Bondo site
where data highlight a positive correlation with the seasonality, with increasing values in summer and
decreasing values in winter, following the trend of both atmospheric pressure and temperature. At
least during the monitored period, at the Bondo site no correlation with local seismicity appears to be
noticeable, while clear correlations between Radon outliers and the rainfall period are evident (Figure
9, panel *a*). It is worth mentioning how not all Radon sensors show the same behaviour with respect
to the season. At the Montelungo site (Table 1), for instance, data show a complete anti-correlation
with the seasons (Figure 9, panel *b*), with lower values in summer and higher values in winter,
probably as a consequence of the different local geological and morphological setting.
A further example regards the $CO_2$ flux variation, measured at the Nirano site (Table 1) in
October 2023 (Figure 10, panel *a*) during an intense period of weak motion earthquakes localized
very close to ZO.PDN10 station, installed in the area of the Salse di Nirano regional park. Starting
from June 2023, the area of Nirano showed an increase in the local seismicity. In the period
2023/06/01 to 2023/11/15, 32 earthquakes with local magnitude ($M_L$) in the range 2.0-3.5 were
recorded with a maximum epicentral distance from Nirano of 30 km. In particular, the strong events
with $M_L$=3.5 occurred on 30[th] October 2023 (04:25:53 UTC). Considering the $CO_2$ time series
recorded at Nirano, and a time period spanning from October 15[th] to November 15[th], it is possible to
highlight the presence of many $CO_2$ data points with values exceeding the limit of +1 standard
deviation (Figure 10, panel *b*), with respect to the average values of the period. Many outliers were
recorded just before and also soon after the $M_L$=3.5 target earthquake. It worth noting that this
evidence should be carefully evaluated also considering other parameters (for instance, atmospheric
pressure, soil moisture and temperature), even though no relevant rainfall episodes occurred in that
period.





Further case studies concern the correlation between meteorological (i.e. temperature and
rainfall) and groundwater parameters. Understanding water level fluctuation patterns is one of the
pillars for designing adaptive management practices that can mitigate the impacts of extreme water
levels on infrastructure and associated economic activities (e.g. Gerten et al., 2013, Alley et al., 2002,
Taylor et al., 2009; Russo and Lall, 2017). Groundwater recharge is difficult to estimate, especially
in fractured aquifers, because of the spatial variability of the soil properties and because of the lack
of data at basin scale. A possible solution consists in inferring recharge directly from the observation
in boreholes (Guillaumot et al., 2022), even if the direct measures in wells overlook the impact of
lateral groundwater redistribution in the aquifer. When evaluating the effect of exogenous parameters
on groundwaters, rainfall is the main factor in promptly influencing all monitored groundwater
parameters (e.g. Mancini et al., 2022, Guillaumot et al., 2022), with a variable aquifer response with
respect comparable amounts of precipitation in the same period. Figure 11 shows an example of
groundwater recordings related to the meteorological event of October 2023 at Volargne and Fonte
sites (table 1). It is worth noting how in the first site (Figure 11, panel *a*) a gradual and moderate rise
in water level is contrasted by a faster and larger (less than 2 hours) increase in water level at the
Fonte site (Figure 11, panel *b*), which shows an intense influence of the rainfall, also proved by the
extremely variable temperature and conductivity records not observed in the other sites (Ferrari et al.,
2024). The light grey box in Figure 11 highlights another instant response of Fonte groundwater to
precipitation which is even more sharper than the one described above and also involves electrical
conductivity decrease and temperature increase.
The atmospheric temperature is moreover proven to affect groundwater temperature,
especially in aquifers down to ~ 20 m depth (e.g. Lee and Hahn, 2006, 2021; Taylor and Stefan, 2009;
Menberg et al., 2014). Monitoring sites having at least 1 year of recordings are taken into account to
analyse groundwater temperature seasonal oscillations and correlations to air temperature. In our
case, the absence of seasonality is detected at the Bondo, Montelungo and Volargne sites (Table 1)
where the constant groundwater temperatures could be explained by the aquifer depth (~ 50 m). In



other cases, such as the Casaglia site (Table 1), despite the depth of aquifer of a few meters below
ground level, the relevant water (~ 40 m) column above the hydrogeochemical sensor dampened
possible temperature fluctuations (Bucci et al., 2020; Egidio et al., 2022). On the contrary, at the
Balconi site (Table1), despite the aquifer depth is greater than 50 m, a nearly seasonal variation
characterized by maximum values reached in summer and minima in winter, is observed, in
agreement with the measured air temperature periodicity (Ferrari et al., 2024). However, it should be
evaluated whether this temperature variation in specific seasons is directly attributable to
meteorological reasons or to anthropogenic causes, due to intense irrigation occurring in the area of
Balconi during the most dry and hot months of the year.

**7 Data Availability**

Data and metadata presented and described in this manuscript can be accessed
under https://doi.org/10.13127/muda (Massa et al., 2023).

**8 Usage Notes and conclusions**

The technical validation allowed us to obtain a reliable and homogenous dataset of continuous
multiparametric time series and associated metadata. For hydrogeochemical, meteorological, Radon
and $CO_2$, data are published in a raw format after a pre-processing whose main scope was just to
detect gaps and spurious peaks to be deleted from the time series. Seismic data are published after
applying a 5-minute resampling to the raw miniseed 24-hour continuous data and then by converting
the velocity ground motion (cm/s) in RMS and FFT discrete time series. The raw seismic waveforms
are however downloadable from the EIDA-Italia webservice (https://eida.ingv.it/it/).
For the first time, at least in Italy, high frequency and continuous multiparametric data are
dynamically updated daily and published soon after for end users. Data can be used for different



purposes, ranging from i) information regarding environmental and meteorological temporal trends
with respect the global climate change problematic; ii) details on local aquifers features and
seismicity; iii) recommendations for the civil protection; iv) multiparametric geophysical,
environmental and geochemical data for research studies. In particular, all seismic stations included
in MUDA-db with code ZO (PDnet, https://eida.ingv.it/it/networks/network/ZO) contribute with the
national seismic monitoring by sharing a continuous data stream in real time to the INGV National
Seismic Network (https://eida.ingv.it/it/networks/network/IV). In particular, the dense ZO network
installed around Lake Garda contribute significantly to improving the minimum magnitude threshold
detection of the area as reported in Ferrari et al. (2024).

Seismic data, together with all geological, morphological and geophysical information

collected at each site included in MUDA, can moreover be used to investigate the site response in
terms of seismic amplification, in particular for sites installed in the central part of the Po Plain, a
deep basin characterized by a significant thickness of incoherent alluvial deposits. Seismic events
recorded at each station can also be used for local investigations into the micro seismicity of the area,
seismic source recognition or to improve the available seismic velocity models at a local scale.

Hydrogeochemical and geochemical data will be used in the framework of a recent agreement

between the National Institute of Geophysics and Volcanology (INGV) and the National System for
Environmental Protection (SNPA, comprising the Regional Environmental Protection Agencies -
ARPA and ISPRA), aimed at gathering information on seismic activity and aquifer/spring status from
various acquisition sources, in some cases reaching a near real time monitoring through the SINAnet
facility (https://www.snpambiente.it/attivita/sistema-informativo-nazionale-ambientale/).

In general, the multiparametric monitoring is the basis by which to understand and identify

possible seismic precursors, an objective not yet achieved in earthquake studies. In particular, the
short-term earthquake forecasting, remains elusive and largely unattained. An effective solution for
such a major issue might be found, in the future, in systematic high frequency and continuous
measurements with multiparametric networks operating over the long term. Owing to the influx from



deep crustal fluids in active tectonic areas, groundwater monitoring may especially be considered a
fundamental tool for investigating pre-seismic signals of rocks undergoing accelerated strain (e.g.
King, 1986; Skelton, A. et al. 2014, Barberio et al., 2017).

**9 Code Availability**

All the procedures to acquire and process data coming from multiparametric remote stations

have been specifically developed for the MUDA project in Bash scripting language, Python and PHP
language, using, when necessary, proprietary API taken from the manufacturers of the installed
remote instrumentation, as detailed in the text. Seismic data are acquired and archived through a
Seiscomp4 (https://www.seiscomp.de/doc/apps/seedlink.html) client, thereby improving the
SeedLink real time data acquisition protocol. Some processing steps on the seismic data are
undertaken        using        the        Seismic        Analysis        Code        (SAC,
https://ds.iris.edu/ds/nodes/dmc/software/downloads/sac/), a software designed for both real time and
off-line seismological analyses of time series data. The MUDA database is developed with MySQL
open source software. The MUDA web portal is developed in PHP and HTML5 languages, and all
data are published under the Creative Commons License CC BY 4.0 licence.

**Team list**

MUDA working group is at present composed by the main authors and:

Marino Domenico Barberio [4], Rodolfo Puglia [1], Santi Mirenna [1], Ezio D'Alema [1], Anna Figlioli
[1], Fabio Varchetta [1], Antonio Piersanti [4], Valentina Cannelli [4], Gianfranco Galli G. [4], Gianfranco
Tamburello [5], Gioia Capelli Ghioldi [5]

[1] National Institute of Geophysics and Volcanology (INGV), Milano department, Italy



[4] National Institute of Geophysics and Volcanology (INGV), Roma1 department, Roma, Italy
[5] National Institute of Geophysics and Volcanology (INGV), Bologna department, Italy

**Author contributions**


This study started from an original idea by MM and ALR. MM, ALR, EF and SL contributed
to all phases of site installations, data acquisition, processing and archiving. DS developed MySQL-
Db and the associated web page, as well as the procedures to upload data into MUDA-db. SL and EF
contributed to organising the technical data sheet related to each multiparametric site now available
at each single station web page. MM developed the procedures for multiparametric data pre-
processing and seismic data post processing. All authors participated in the preparation of the
manuscript draft.

**Competing Interests: The authors declare no competing interests.**


**Acknowledgements**


We would like to thank all those who contributed to the site search phase and the installation
of the PDnet network. In particular: Prof. Tullia Bonomi (Milano Bicocca University), Engr. Michela
Biasibetti and Engr. Bruno Pannuzzo (Acque Bresciane company), Engr. Massimo Carmagnani and
Engr. Ignazio Leone (Acque Veronesi company), Engr. Giovanni Lepore and Fabrizio Brunello
(Azienda Gardesana Servizi company), Engr. Paolo Pizzaia (Alto Trevigiano Servizi company), Arch.
Luca Bertanza (municipality of Tremosine Garda, BS), Arch. Umberto Minuta (municipality of
Dolcè, VR), Geom. Elena Beraldini (municipality of Negrar, VR), Dr. Marzia Conventi (municipality
of Fiorano Modenese, MO - Dir. Riserva Salse di Nirano), Mr. Salviani (Norcia), Bagni Triponzo
Terme S.p.A. (municipality of Cerreto di Spoleto), Dr. Luca Martelli and Dr. Paolo Severi (Emilia





Romagna Region). Particular thanks go to OGS (National Institute of Oceanography and Applied
geophysics, https://www.ogs.it/en) and University of Ferrara (Department of Physics and Earth
Science, http://fst.unife.it/en) for sharing data recorded at the TOPPO site in the framework of the
ECCSEL-ERIC (European Research Infrastructure for $CO_2$ Capture, Utilisation, Transport and
Storage (CCUS), https://www.eccsel.org/) consortium.
The production of the seismic data published in the MUDA database involves many INGV colleagues
covering different tasks in the data production chain, from the correct operation of the seismic stations
to data acquisition, data processing, archiving and subsequent distribution. We would like to thank all
the colleagues of the INGV sections and offices who contribute daily to the management of the
National Seismic Network (RSN, https://eida.ingv.it/it/networks/network/IV), as well as providing
access to the data they produce.

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

**Figure Captions**

**Figure 1 -** MUDA home page: https://muda.mi.ingv.it. Examples of interactive pop-ups are reported
in map indicating the available main options concerning stations data and metadata, Italian seismic
hazard (Stucchi et al., 2011), seismicity rate and both composite and single seismogenic sources
(DISS database, DISS Working Group 2021). The base map is provided by ©OpenStreetMap
contributors 2024. Distributed under the Open Data Commons Open Database License (ODbL) v1.0.

**Figure 2** - Target areas and relative multiparametric sites. Each panel indicates the multiparametric
site (yellow triangles), the Italian seismic hazard map in terms of horizontal peak ground acceleration
(PGA) with 10% probability of exceedance in 50 years on hard ground (Stucchi et al., 2011), the
instrumental seismicity from 1985 (black circles, https://terremoti.ingv.it), the historical seismicity
(red circles, CPTI database, Rovida et al., 2020) and the seismogenic sources (CSS-DISS database,
DISS Working Group 2021). The base maps are provided by ©OpenStreetMap contributors 2024.
Distributed under the Open Data Commons Open Database License (ODbL) v1.0.

**Figure 3** - Dynamic multiparametric data web page:  https://muda.mi.ingv.it/dat.php. From top to
bottom: hydrogeochemical data (light blue panels: water level, m; water electrical conductivity,
$\mu$S/cm; water temperature, °C), meteorological data (green panel: soil temperature, °C; rain, mm),
seismic data (light brown panels: RMS, ground motion velocity, cm/s; FFT, cm/s/Hz; FFT($f$),
cm/s/Hz), $CO_2$ data (grey panels: soil flux,  g*m$^{-2}$*d$^{-1}$; humidity, %; soil temperature, °C), Radon data
on air (yellow panel: Bq/m$^3$). All time series (*csv* format) and each single graph (*pdf*, *png* formats)
are downloadable by using the menu available in the top right corner of each panel.




**Figure 4** - Multiparametric sites web page: https://muda.mi.ingv.it/stazione.php. The single site web
page indicates the main features of both instrumentation and installation, downloadable thematic
maps such as the geological map (1:100.000, Società Geologica Italiana
http://www.isprambiente.gov.it/it/cartografia/), the topographic map (base at 1:25.000, Istituto
Geografico Militare, http://www.igmi.org/prodotti/cartografia/carte_topografiche/), log and
stratigraphy concerning the available wells for water, a preliminary geophysical soil characterization
in term of horizontal to vertical spectral ratio performed on ambient seismic noise and the complete
time hydrogeochemical time series. Sources for base maps: Esri, DigitalGlobe, GeoEye, i-cubed,
USDA FSA, USGS, AEX, Getmapping, Aerogrid, IGN, IGP, swisstopo, and the GIS User
Community.

**Figure 5 -** MUDA data base scheme and processing flow chart.

**Figure 6** - Single station data set for multiparametric sites having at least 6 months of data and three
different type of acquisition. S: seismic data (red); Id: hydrogeochemical data (blue); M:
meteorological data (green); R: Radon data (grey); C: $CO_2$ data (yellow). Yellow and orange starts
indicate recorded earthquakes at each site with magnitude (Moment or Local) lower and higher then
4, respectively.

**Figure 7** - Quality check of seismic data at Oppeano site (table 1).
Panel *a* (top): timeseries recorded on December, 29, 2020, by IV.OPPE station
(https://terremoti.ingv.it/instruments/station/OPPE) showing consecutive earthquakes: the first, with
$M_w$=6.3, occurred in Croatia land (https://terremoti.ingv.it/event/25870121) 450 km East of IV.OPPE
and the others with epicentres 11 km South-West of IV.OPPE with maximum $M_w$=3.9



(https://terremoti.ingv.it/event/25871441). Earthquakes origin times (UTC) are reported in the top
panel *a*).
The bottom panel *a*) shows the FFT functions calculated considering 15 frequency intervals from 0.1
to 20 Hz, considering 288 consecutive 5 minutes-time windows (i.e. 24 hours) selected on the vertical
component of motion.
Panel *b*: from top to bottom, the RMS, the ground motion velocity and the mean FFT calculated for
288 consecutive time windows with duration of 5 minutes. Red, blue and green indicate the vertical,
the North-South and the East-West horizontal components of motion. The black solid lines indicate
the cumulative functions.

**Figure 8** - Example of multiparametric data recorded at Bondo site (table 1). Panel *a* shows the
hydrogeochemical data recorded in the time period from October, 15 to November, 15, 2023: black,
red and green solid lines indicate the water level (m), the electrical conductivity (µS/cm) and the
water temperature (°C) variations, respectively.
The panel *b* shows the seismic data recorded at ZO.PDN3 station (table 2) on October, 31 and
November, 1, 2023, while the panel *c* shows the polarization analysis in terms of rotated horizontal
to vertical spectral ratio.

**Figure 9 -** Radon time series (Bq/m$^3$, black lines) recorded at Bondo (panel *a*) and Montelungo (panel
*b*) multiparametric sites. Soil temperature (°C, green lines) and rain (mm, light blue lines) are also
indicated. At the top of each panel seasons are indicated (Aut=Autumn, Win=Winter, Spr=Spring,
Sum=Summer).

**Figure 10** - Panel *a*: CO$_2$ flux (g*m$^{-2}$*d$^{-1}$, black line) recorded at NIRANO multiparametric site. Red
and green solid lines indicate the smoothing function of CO$_2$ flux and the temperature (°C),
respectively. The orange box indicates the time window represented in the bottom panel.





Panel *b*: detailed monitoring for period October, 25, 2023 to November, 15, 2023, where a seismic
sequence, with maximum local magnitude of 3.5 (vertical yellow dashed lines) occurred in
correspondence of the Nirano's Mud-Volcanoes. Dotted and dashed orange lines indicate the mean
values +/- 1-standard deviation of $CO_2$ flux recorded during the analysed time period. Rain values
(mm, light blue) and soil temperature (°C) are also indicated. Black dots indicate the $CO_2$ measures
($g*m^{-2}*d^{-1}$).

**Figure 11** - Groundwater recharge at Volargne (panel *a*) and Fonte (panel *b*) sites (table 1) recorded
on October, 30, 2023. In both panels, black, red and green solid lines indicate the water level (m), the
electrical conductivity (µS/cm) and the water temperature (°C) variations, respectively, while the blue
solid lines indicate the cumulative rain.

**Table captions**

**Table 1** - Monitoring multiparametric sites at present include in MUDA.

**Table 2** - Main features of hydrogeochemical probes and seismic monitoring stations. Grey cells
indicate well with thermalism (i.e. Triponzo site), while the black cells indicate the borehole seismic
sensor installed 150 m depth at Oppeano site.





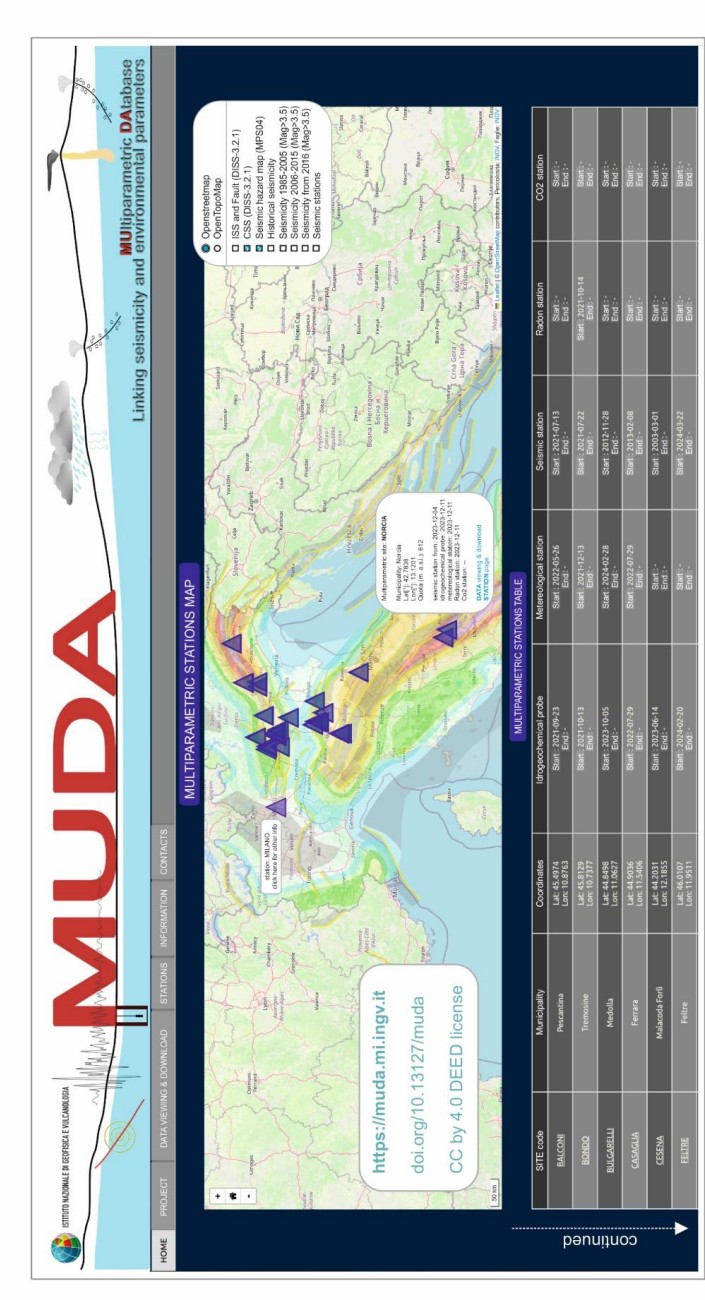


Figure 1



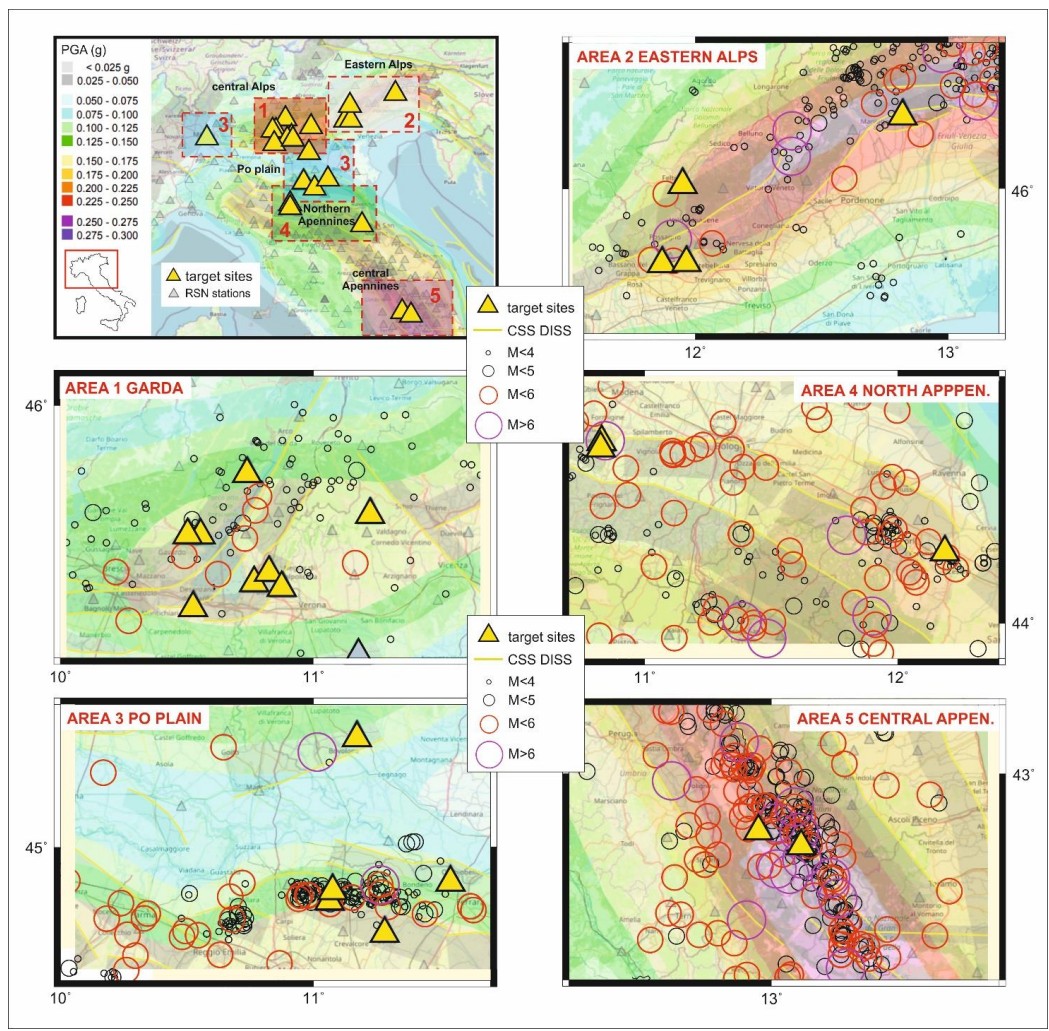



Figure 2











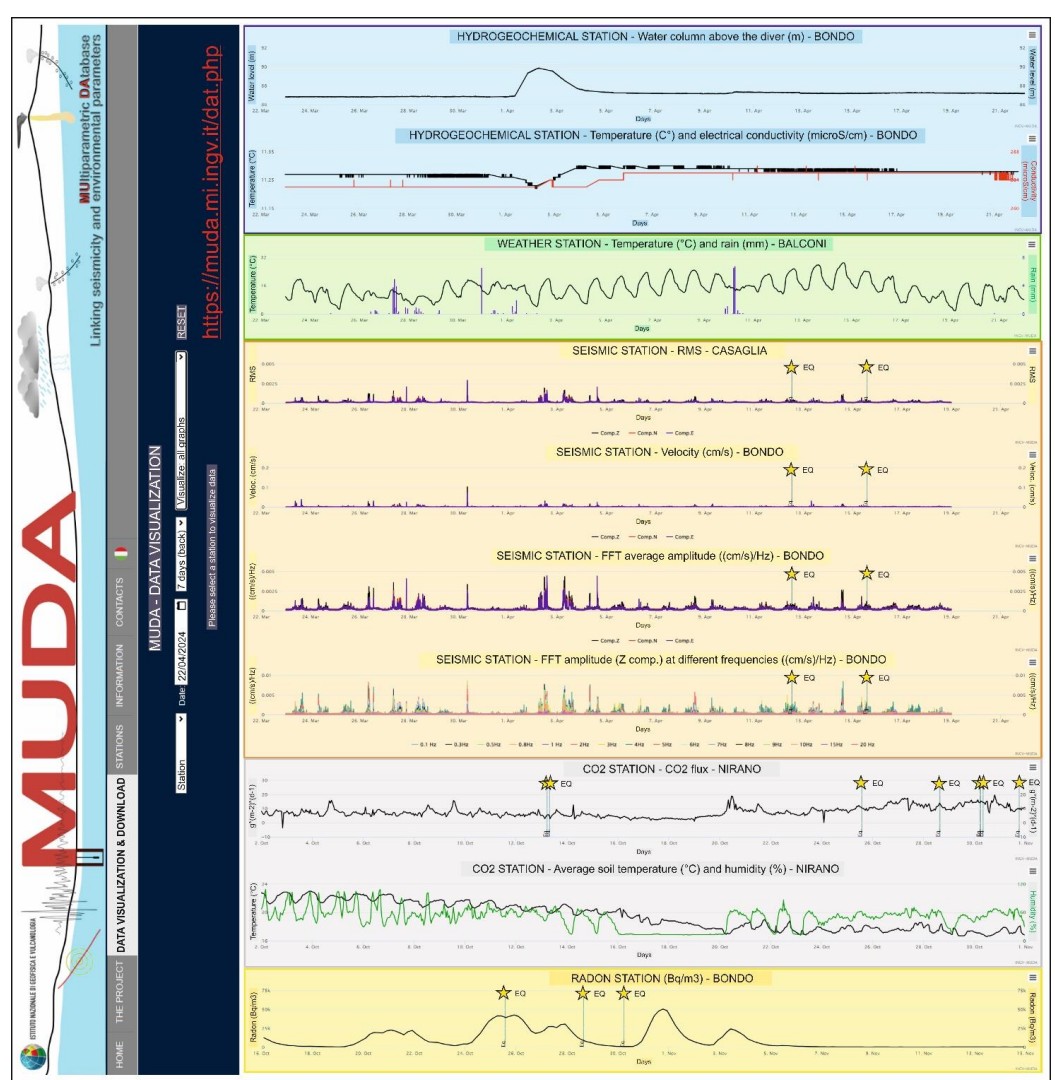



Figure 3












Figure 4




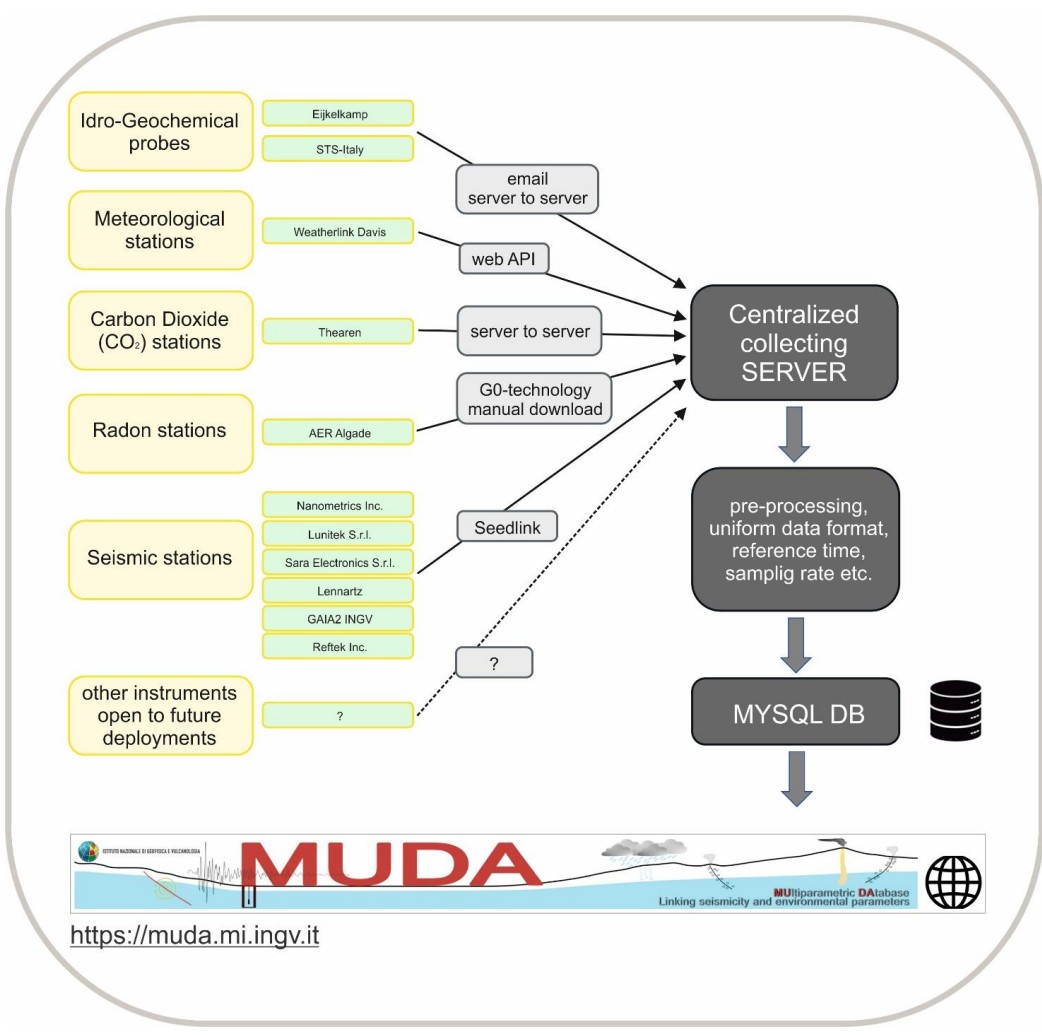



Figure 5











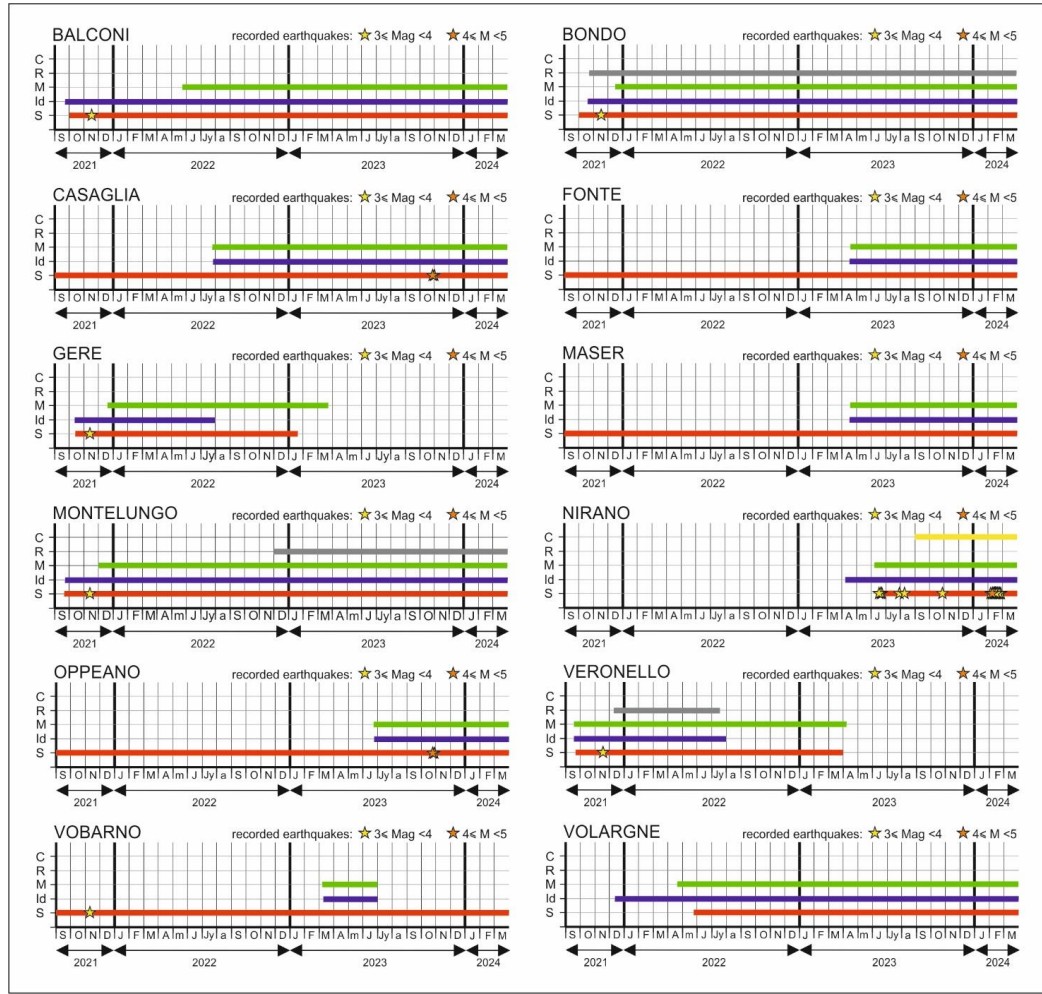



Figure 6













Figure 7



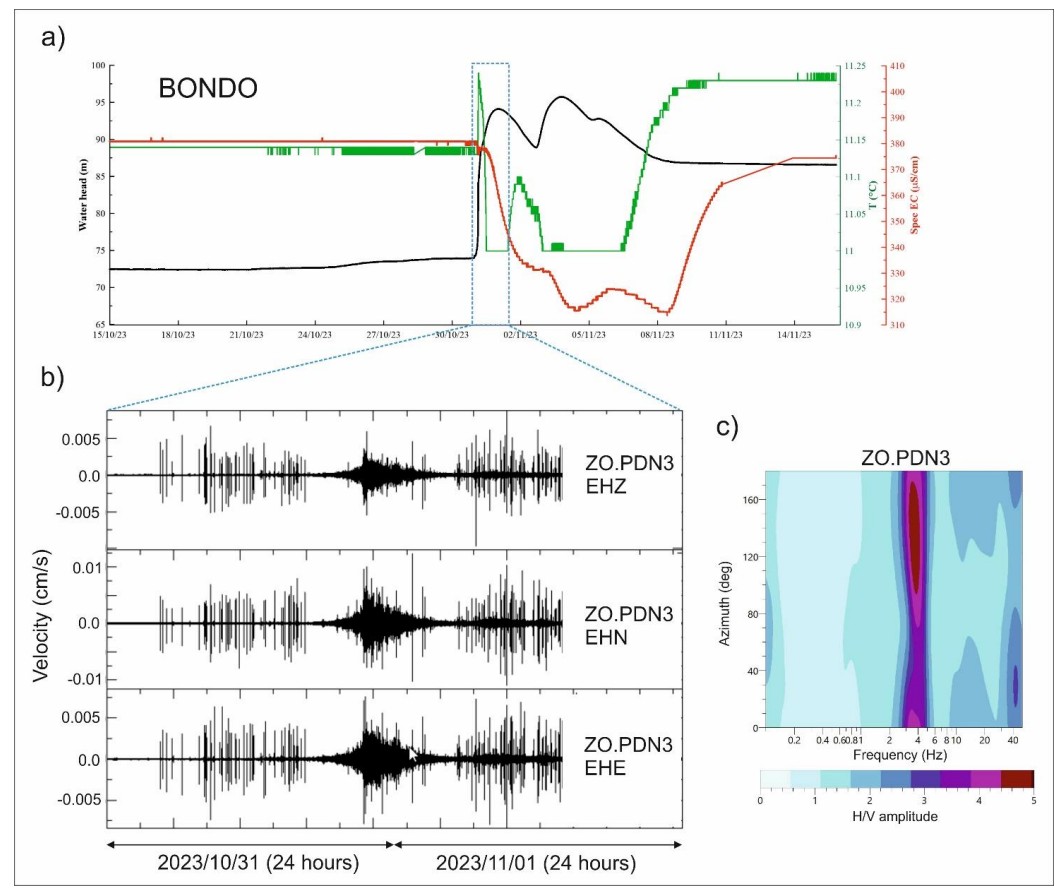



Figure 8











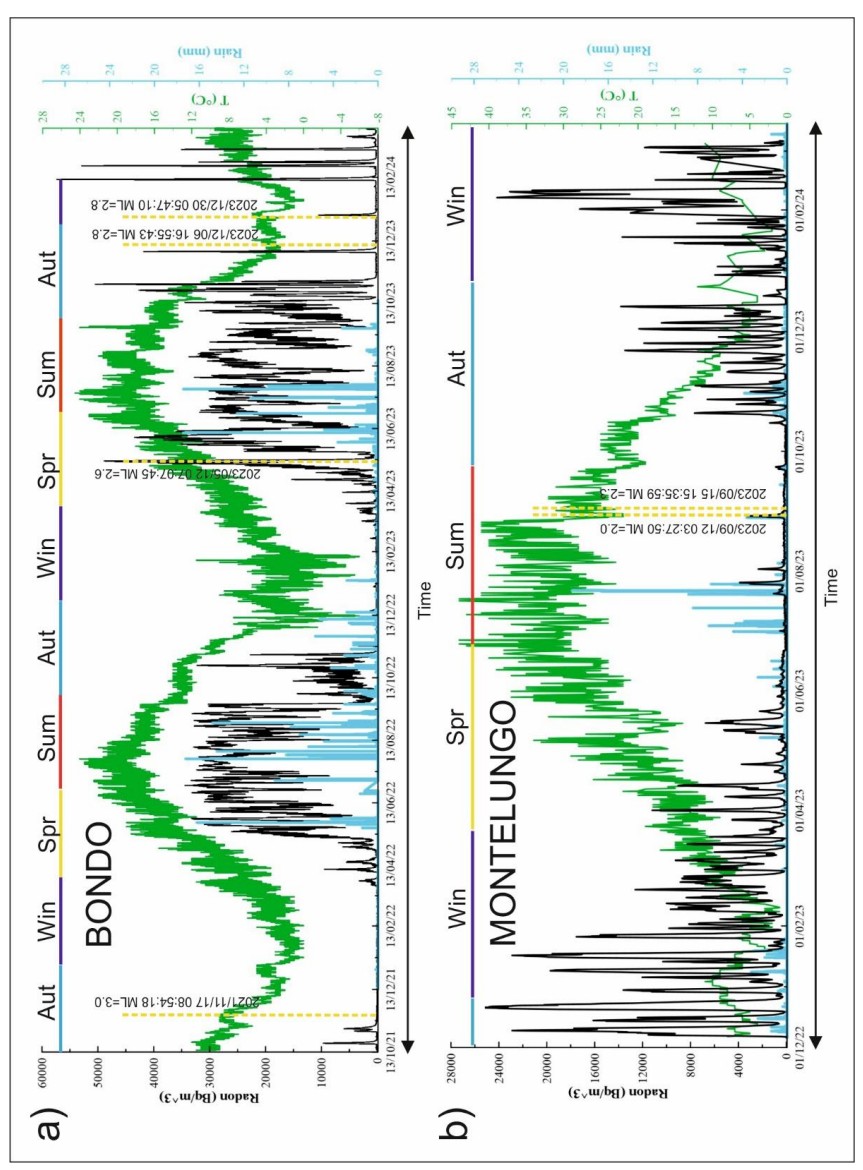



Figure 9









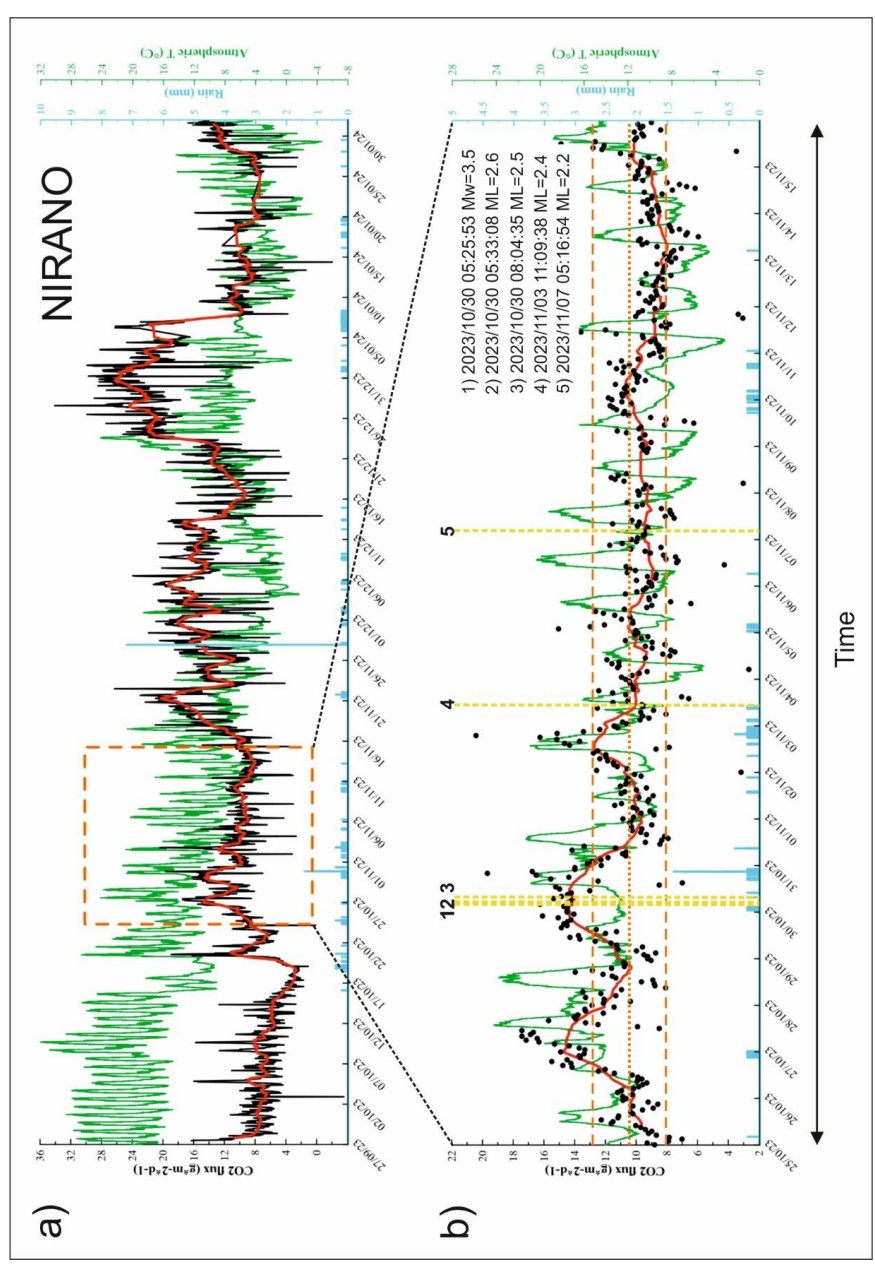



Figure 10





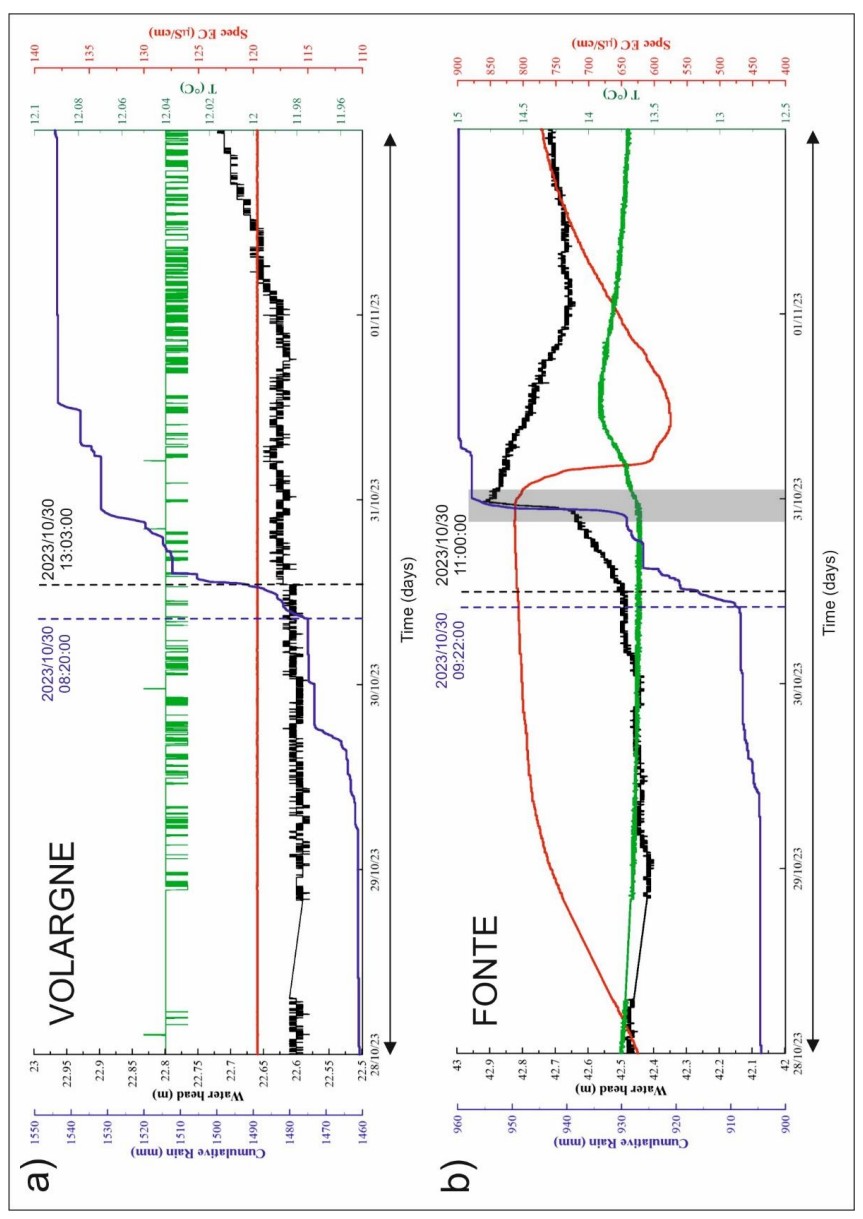



Figure 11





| CODE | Municipality | AREA | LAT [°] | LON [°] | idro-geochemical | weather | seismic | Radon | CO2 |
|---|---|---|---|---|---|---|---|---|---|
| BALCONI | Pescantina | 1 | 45.4974 | 10.8763 | start 2021-09-23 | start 2022-05-26 | start 2021-07-13 | / | / |
| BONDO | Tremosine | 1 | 45.8129 | 10.7377 | start 2021-10-13 | start 2021-12-13 | start 2021-07-22 | start 2021-10-14 | / |
| BULGARELLI | Medolla | 3 | 44.8498 | 11.0627 | start 2024-02-28 | start 2024-02-28 | start 2012-11-28 | / | / |
| CASAGLIA | Ferrara | 3 | 44.9036 | 11.5406 | start 2022-07-29 | start 2022-07-29 | start 2013-02-08 | / | / |
| CESENA | Malacoda Forlì | 4 | 44.2031 | 12.1855 | start 2024-02-20 | / | start 2003-03-01 | / | / |
| FELTRE | Feltre | 2 | 46.0107 | 11.9511 | start 2024-02-20 | / | / | / | / |
| FONTE | Fonte | 2 | 45.7949 | 11.8697 | start 2023-04-18 | start 2023-04-17 | start 2011-11-17 | / | / |
| GERE | Gardone Riviera | 1 | 45.6422 | 10.5484 | start 2021-10-12 - end 2022-07-30 | start 2021-12-02 - end 2023-03-25 | start 2021-10-12 - 2023-01-18 | / | / |
| MASER | Maser | 2 | 45.7969 | 11.9658 | start 2023-04-17 | start 2023-04-17 | start 2011-11-17 | / | / |
| MEDOLLA | Medolla | 3 | 44.8492 | 11.0734 | start 2024-02-20 | start 2024-02-28 | start 2012-11-28 | / | / |
| MILANO | Milano | 3 | 45.4972 | 9.1812 | start 2024-03-12 | start 2024-03-12 | start 2012-01-27 | / | / |
| MIRANDOLA | Mirandola | 3 | 44.8812 | 11.0782 | start 2024-02-23 | start 2024-02-23 | start 2012-11-28 | / | / |
| MONTELUNGO | Desenzano del Garda | 1 | 45.4429 | 10.5256 | start 2021-09-14 | start 2021-11-30 | start 2021-07-30 | start 2022-12-06 | / |
| NIRANO | Fiorano Modenese | 4 | 44.5141 | 10.8255 | start 2023-04-04 | start 2023-06-06 | start 2023-06-23 | / | start 2023-09-27 |
| NIRANO1 | Fiorano Modenese | 4 | 44.5002 | 10.8163 | start 2024-02-20 | start 2024-02-20 | start 2023-06-23 | / | / |
| NORCIA | Norcia | 5 | 42.7838 | 13.1201 | start 2023-12-11 | start 2023-12-11 | start 2023-12-04 | start 2023-12-11 | / |
| OPPEANO | Oppeano | 3 | 45.3082 | 11.1723 | start 2023-06-29 | start 2023-06-21 | start 2023-02-16 | / | / |
| RECOARO | Recoaro Terme | 1 | 45.6998 | 11.2217 | start 2024-01-30 | start 2024-01-30 | / | / | / |
| RECOARO1 | Recoaro Terme | 1 | 45.7005 | 11.2215 | start 2024-01-30 | start 2024-01-30 | / | / | / |
| RENAZZO | Renazzo | 3 | 44.7624 | 11.2836 | start 2024-02-20 | / | start 2003-10-11 | / | / |
| SECCHIA | Concordia Secchia | 3 | 44.9245 | 11.0183 | start 2024-03-14 | start 2024-03-20 | start 2012-11-28 | / | / |
| TOPPO | Toppo di Travesio | 2 | 46.1985 | 12.8171 | start 2024-02-25 | / | / | / | / |
| TRIPONZO | Triponzo | 5 | 42.8400 | 12.9480 | start 2023-12-12 | start 2023-12-12 | start 2023-12-04 | start 2023-12-12 | / |
| VERONELLO | Bardolino | 1 | 45.5098 | 10.7645 | start 2021-09-14 - end 2022-08-01 | start 2021-10-19 - end 2023-04-07 | start 2021-07-08 - end 2023-04-07 | start 2021-12-17 - end 2022-07-08 | / |
| VOBARNO | Vobarno | 1 | 45.6428 | 10.5035 | start 2023-03-10 - end 2023-06-21 | start 2023-03-10 - end 2023-06-21 | start 2021-07-08 | / | / |
| VOLARGNE | Dolcè | 1 | 45.5397 | 10.8235 | start 2021-12-16 | start 2022-05-20 | start 2022-04-12 | start 2023-12-12 | / |

Table 1





Table 2

| CODE | AREA | Geology (100k) | Topography | SITE TYPE | WELL DEPTH (m) | WATER LEVEL (m) | WATER COLUM (m) | SEISMIC NET | SEISMIC CODE | RECORDER | SENSOR |
|---|---|---|---|---|---|---|---|---|---|---|---|
| BALCONI | 1 | fluvio-glacial deposits | plain | well | 100 | 53.3 | 46.7 | ZO | PDN2 | Lunitek ATLAS | TELLUS-5s |
| BONDO | 1 | alluvial deposits | valley | well | 180 | 44 | 136 | ZO | PDN3 | Reftek-130 | LENNARTZ-5s |
| BULGARELLI | 3 | alluvial deposits | plain | well | 8 | 1.5 | 6.5 | IV | CAVE | GAIA2 | TRILLIUM-120s |
| CASAGLIA | 3 | alluvial deposits | plain | well | 130 | 3.5 | 126.5 | IV | FERS | GAIA2 | TELLUS-5s |
| CESENA | 4 | alluvial deposits | plain | well | 8.3 | 3.1 | 5.2 | IV | BRSN | GAIA2 | LENNARTZ-1s |
| FELTRE | 2 | limestone | relief | spring | / | / | / | / | / | / | / |
| FONTE | 2 | sandstone | relief | well | 120 | 7.7 | 112.3 | IV | ASOL | Lunitek ATLAS | TELLUS-5s |
| GERE | 1 | fluvio-glacial deposits | valley | well | 60 | 31.3 | 28.7 | ZO | PDN6 | Lunitek ATLAS | TELLUS-5s |
| MASER | 2 | sandstone | relief | well | 157 | 71.3 | 85.7 | IV | ASOL | Lunitek ATLAS | TELLUS-5s |
| MEDOLLA | 3 | alluvial deposits | plain | well | 50 | 3 | 47 | IV | CAVE | GAIA2 | TRILLIUM-120s |
| MILANO | 3 | alluvial deposits | plain | well | 152 | 18 | 135 | IV | MILN | GAIA2 | TRILLIUM-40s |
| MIRANDOLA | 3 | alluvial deposits | plain | well | 300 | 5.4 | 294.6 | IV | CAVE | GAIA2 | TRILLIUM-120s |
| MONTELUNGO | 1 | morainic deposits | hill | well | 150 | 52.8 | 97.2 | ZO | PDN4 | Reftek-130 | LENNARTZ-5s |
| NIRANO | 4 | mudstone | hill | mudhole | 11 | 0 | 11 | ZO | PDN10 | GAIA2 | LENNARTZ-5s |
| NIRANO1 | 4 | mudstone | hill | mudhole | 11 | 0 | 11 | ZO | PDN10 | GAIA2 | LENNARTZ-5s |
| NORCIA | 5 | alluvial deposits | valley | well | 64 | 30 | 34 | ZO | PDN11 | Reftek-130 | LENNARTZ-5s |
| OPPEANO | 3 | alluvial deposits | plain | well | 60 | 4.6 | 55.4 | ZO | PDN9 | SARA-SL06 | SARA-SS08-120s |
| RECOARO | 1 | sandstone | relief | spring | / | / | / | ZO | PDN13 | Lunitek ATLAS | TELLUS-5s |
| RECOARO1 | 1 | sandstone | relief | spring | / | / | / | ZO | PDN13 | Lunitek ATLAS | TELLUS-5s |
| RENAZZO | 3 | alluvial deposits | plain | well | 6 | 1.5 | 4.5 | IV | RAVA | GAIA2 | LENNARTZ-5s |
| SECCHIA | 3 | alluvial deposits | plain | mudhole | 1 | 0 | 1 | IV | CAVE | GAIA2 | TRILLIUM-120s |
| TOPPO | 2 | alluvial deposits | valley | well | 300 | 114 | 186 | / | / | / | / |
| TRIPONZO | 5 | limestone | crest | well | 53 | 3.1 | 49.9 | ZO | PDN12 | Reftek-130 | LENNARTZ-5s |
| VERONELLO | 1 | fluvio-glacial deposits | hill | well | 198 | 90 | 108 | ZO | PDN1 | Lunitek ATLAS | TELLUS-5s |
| VOBARNO | 1 | alluvial deposits | valley | well | 36 | 11.2 | 24.8 | IV | VOBA | Lunitek ATLAS | TELLUS-5s |
| VOLARGNE | 1 | fluvio-glacial deposits | valley | well | 99 | 50 | 49 | ZO | PDN8 | Reftek-130 | LENNARTZ-5s |