# Peer review of "MUDA: dynamic geophysical and geochemical MUltiparametric DAtabase"

_Earth System Science Data, 2024_

## Referee Comment (RC1)

**Review of the manuscript No. essd-2024-185 '*MUDA: dynamic geophysical and geochemical MUltiparametric DAtabase*' submitted to *Earth System Science Data (ESSD).***
* * *
Recommendation: accept.

Focus of the paper: this is the data descriptor article where the authors presented the new dynamic geophysical and geochemical MUltiparametric DAtabase (MUDA).

Relevance: The presented study is the original primary research within the scope of the journal since it presents a novel software for seismic and geophysical data processing.

Abstract is well written and clearly describes the undertaken study.

Structure: The article is well organized with structured sections and subsectiocs. The logic between these sections is well maintained.

Introduction presents a background, defines research goals and provides a clear statement of research problem of processing and handling seismic data efficiently. The authors described the existing main seismic networks of Italy: RSN managed by INGV, Federation of Digital Seismograph Networks, FDSN, International Seismological Center, ISC, SEED etc. The state-of-the-art section presents the purpose of the research investigation which is supported by the pertinent literature. Literature is well referenced and relevant.

Motivation is explained: today, the interaction between tectonics and crustal fluids dynamics is still lacking a simultaneous monitoring of the relative key factors. This study contributes to fill in the gaps through presenting a novel instrument for development of multiparametric monitoring (MUDA).

English language: acceptable. Clear, unambiguous, professional English language used throughout.

Data used in this study are described: The authors used data collected by multidisciplinary monitoring networks stored in MUDA (geophysical and geochemical MUltiparametric DAtabase), a new dynamic multiparametric database.

Methods: The methodological approach is well explained and described: MUDA is a MySQL relational database with a web interface developed in php, aimed at investigating in quasi real time possible correlations between seismic phenomena and variations in endogenous and environmental parameters. Methods described with sufficient information. The workflow is well structured.

Results are reported: The authors present case examples of using data from MUDA using five main target areas, such as the Garda region, Lake Garda, eastern Alps, Po alluvial basin and Northern and Central Apennine chains. These data enabled to test and compare seismicity in these areas. They well explain how the data are applied and used. Thus, the authors noted functional approach of MUDA: it collects data from different types of sensors such as hydrogeochemical probes for physical-chemical parameters in waters, meteorological stations, detectors of air Radon concentration, diffusive flux of carbon dioxide ($CO_2$) and seismometers belonging both to the National Seismic Network of INGV and to temporary networks installed in the framework of multidisciplinary research projects.

Discussion interpreted the major outcomes of this study: the authors highlighted the efficiency of MUDA which daily publishes data updated to the previous day and offers the chance to view and download multiparametric time series selected for different time periods. The advantages of the results are described regarding the novel tool for network data management and seismic data sharing.

Conclusion The resultant dataset provides broad perspectives in the framework of future high frequency and continuous multiparametric monitoring. The importance of this paper is well summarized as follows: MUDA contributes to the development of seismic monitoring tools aimed at detecting and identification of possible seismic precursors for short-term earthquake forecasting. This clearly contributes to the mitigating geological hazard and risks in

mountainous areas such as Italy.

Actuality, novelty and importance of the research is clear: The authors developed MUDA – a new infrastructure of the National Institute of Geophysics and Volcanology (INGV) which aims at archiving and disseminating multiparametric seismic data

Academic contribution: The paper increases the knowledge in seismic monitoring, mitigating geophysical hazards and risks through operative monitoring of seismic signals using MUDA. The paper deserved to be published in ***Earth System Science Data (ESSD)***.

Figures: Figures are of acceptable quality, easy to read, relevant and suitable. They include seismograms, print screens of MUDA, with working interface, tables, maps and workflow chart, and other illustrations

Recommendation: This manuscript can be accepted based on the detailed report above.

With kind regards,

- Reviewer.

13.07.2024.

---

## Referee Comment (RC2)

Review of the manuscript by Galina Kopylova
Title: MUDA: dynamic geophysical and geochemical MUltiparametric DAtabase
Author(s): Marco Massa, Andrea Luca Rizzo, Davide Scafidi, Elisa Ferrari, Sara Lovati, Lucia Luzi, and the MUDA working group
MS No.: essd-2024-185
MS type: Data description paper
Iteration: Initial submission

General comments

The manuscript describes a new dynamic geophysical and geochemical MUltiparametric DAtabase (MUDA), created at the National Institute of Geophysics and Volcanology (INGV) and published online in December 2023.

MUDA is based on a MySQL relational database with a web interface developed in PHP. In addition, the authors use additional processing programs written in PITON.

The purpose of MUDA is the automated collection, storage, archiving of multi-parameter data collected by multidisciplinary monitoring networks in Italy, and providing interactive user access to information about observation stations and to the received data via the Internet (https://muda.mi.ingv.it/index.php?lang=en).

The specified link provides access to DAtabase (MUDA), which allows you to familiarize yourself with the details of the creation of the project, with an interactive map of stations, data on all stations. This made it possible to obtain a full understanding of the capabilities of MUDA and compare them with the text of the manuscript.

Data on stations includes information on the geological and geomorphological conditions of their location, coordinates, and a brief description of wells and equipment. It is possible to view data for the entire observation period and at the latest time intervals of 1 day, week, month. It is possible to download data in cvc format and in the form of time graphs.

Of greatest interest to me was assessing the possibility of studying correlations between the behavior of hydrogeochemical data (level/pressure, temperature and electrical conductivity of groundwater in wells, concentrations of radon and carbon dioxide in the air) with endogenous and exogenous factors.

By endogenous factors, the authors apparently mean seismic signals recorded at a network of seismic stations and earthquake catalogs. Exogenous factors apparently refer to data from meteorological stations measuring atmospheric pressure, air temperature and precipitation.

Unfortunately, many shortcomings of MUDA for solving this stated problem have been revealed.

Specific comments

1. An analysis of the possibility of using MUDA to study the correlation between hydrogeochemical data, seismicity and meteorological parameters revealed the following disadvantages that do not allow us to effectively solve this problem:

- the text lacks a general description of the well network, in particular, there is no data on the range of depths of observation wells, temperature, salinity and chemical composition of water;

- time series of observations at wells are predominantly short and limited to the first years (what is the reason?);

- there is no information about the technogenic influence on the behavior of hydrogeochemical parameters, although the factor of technogenic influence in urbanized areas is known to have a significant impact, especially on groundwater in shallow aquifers;

- there is no information on the quality of observation data on groundwater parameters in wells: from the overview graphs of observational data for all time, it follows that the database contains both registration data and data representing technical defects; analysis of data quality for the entire observation period for individual wells is missing and, apparently, was not provided for;

- there is no information about logs of visits to individual wells and equipment maintenance work.

The authors of MUDA carried out additional processing of the original seismic records with a frequency of 100 and 200 Hz in order to ensure their loading and viewing in MUDA. In my opinion, this is an unnecessary procedure that does not have a scientific basis for solving problem No. 1. Data from earthquake catalogs would be sufficient to solve it.

2. Similar remarks arise when solving the problem of searching for correlations between the behavior of geochemical parameters - radon and carbon dioxide with seismicity and meteorological parameters. The results of this analysis also showed that there are very few geochemical observation stations and corresponding data. This should also be written about in the text.

3. The presentation of graphical material in the manuscript and in MUDA is mostly unsatisfactory:
- small, unclear font makes it difficult to see dates;
- on the Time scale, years are not indicated;
- there are errors in the signatures (Fig. 4, 5 – "Idrogeochemical???";
- the dates of inspection of well bores are not indicated.
Graphs are presented rotated 180 degrees.
It is generally very difficult to work with a manuscript, since the text, illustrations and captions are located in different places.

Conclusion
Despite all these shortcomings, MUDA works and shows an example of creating an automated system of multiparametric data, with the help of which current scientific problems can be solved in the future, including the study of correlations between seismic phenomena, changes in groundwater parameters in wells, taking into account meteorological data.

MUDA currently publishes daily data updated from the previous day and offers the ability to view and download multi-parameter time series selected for different time periods.

The use of MUDA in scientific research at the current stage of its development is problematic due to the lack of a critical analysis of the quality of data in the database.

2. I support the development of MUDA for future multi-parameter monitoring to identify possible short-term precursors of earthquakes. However, for this, the authors need to pay more attention to the features of the observation network and the conduct of observations in wells, as well as to the patterns of behavior of hydrogeochemical parameters and to issues of technogenic influence.

August 5, 2024

---

## Author Response (AR1)

**Reviewer 1: Dr. Polina Lemenkova (no modifications have been requested)**

Dear Dr. Lemenkova,

thank you for the attention to our work and for your careful review.

Best regards, Marco Massa

**Reviewer 2: Dr. Galina N. Kopylova**

Dear Dr. Kopylova,

thank you for the attention to our work and for your careful review. Concerning the specific comments, we indicate below the related answers (in bold):

1) An analysis of the possibility of using MUDA to study the correlation between hydrogeochemical data, seismicity and meteorological parameters revealed the following disadvantages that do not allow us to effectively solve this problem: the text lacks a general description of the well network, in particular, there is no data on the range of depths of observation wells, temperature, salinity and chemical composition of water.

**Thank you for suggestion. In the revised version of the paper we have added a description of temperature and electrical conductivity ranges. We have no data on water salinity, but available data on water electrical conductivity are typically used as a proxy measure of salinity. Temperature and electrical conductivity information have also been added in Table 2. Water chemical analyses are not reported as they have not yet been carried out for all sites, even if in the framework of future projects, a thorough chemical characterisation is planned. As soon as we have further homogeneous analyzes available for all sites, they will be inserted into the database and made available on the MUDA web page.**

2) - time series of observations at wells are predominantly short and limited to the first years (what is the reason?)

**As explained in the manuscript, MUDA arose from the need to archive and distribute multiparametric data after the installation of a multiparametric network in Northern Italy (Ferrari et al., 2024), which began in September 2021. This network includes probes for water, seismic stations, and other types of instrumentation such as Radon, CO2, and meteorological stations. Therefore the longest time series available and archived in MUDA have a duration of (about) 3 years. After the first publication of the web site (December 2023) other projects chose to share their data by MUDA: in this case, however, the available data time series are even shorter (in many cases less than one year). At present MUDA archives and distributes all Italian multiparametric data recorded in continuous mode with high frequency monitoring. In any case, all these information are reported in table 1, where all installation date for each type of instrumentation is clearly shown.**

3) there is no information about the technogenic influence on the behaviour of hydrogeochemical parameters, although the factor of technogenic influence in urbanized areas is known to have a significant impact, especially on groundwater in shallow aquifers.

**Thank you for suggestion. Observations on technogenic influence exerted on hydrogeochemical parameters have been added in the revised version of the manuscript. However, in our opinion, an accurate statistical analysis focusing on the discrimination between natural and anthropic effects is not the focus of the paper, even if it is a very interesting issue. A careful example is reported in the paper of Ferrari et al. (2024) concerning the site of Balconi. In our opinion, at this level the mission of MUDA is to provide accurate raw data: in this way as highlighted in the data processing described in the manuscript, in case of unusual and abrupt peaks (spurious spikes) due to technical problems, the peaks are always removed. Data, on the contrary, are not filtered for possible technogenic disturbances, such as pumping etc., because of they represent intrinsic features of the selected wells or sites of installation. This operation, in our opinion, has to be a part of a further step of processing in dependence of the interest of each single researcher. However, we agree that the knowledge of a priori possible technogenic influences is a very important information, so that, for completeness, in the revised version of table 2 we indicate the presence of pumps in the selected wells.**

4) there is no information on the quality of observation data on groundwater parameters in wells: from the overview graphs of observational data for all time, it follows that the database contains both registration data and data representing technical defects; analysis of data quality for the entire observation period for individual wells is missing and, apparently, was not provided for.

**The paper includes an ad-hoc paragraph devoted to the data quality check describing the data pre-processing in order to provide corrected raw data. In detail, as reported in the manuscript, all different type of data are, at first, resampled in order to simplify the viewing and comparison; then, as described in the text, the preprocessing starts involving (if necessary) the barometric compensation to account for atmospheric pressure variations in order to provide for each probe the corrected water level value. Moreover, all multiparametric data are daily checked for availability and gaps usually due (as example) to possible lack of data transmission platforms during rainy and stormy days or other malfunctionings. Details about the period of data availability are marked in the graphs of figure 6, even if just for the longest time series (for very short period the analysis has no statistical meaning). In the manuscript, it is specified as the recorders, thanks to their internal memory and datalogging capacity, are able to archive data up to a maximum of 30 days in case of lack in the transmission. Finally, the manuscript highlights as in case of unusual abrupt peaks (spurious spikes) due to several problems, the peaks are always removed (we now better specify this aspect in the revised version). On the contrary, as reported above, on purpose, data are not correct for technogenic influences in case they are an intrinsic feature of the selected wells (i.e. possible pumping, summer irrigation etc.) because in our opinion data have to be provided in the raw (corrected) format. As example, the filtering for pumping or others anthropogenic sources has to be a further and deepened step of processing in charge to the end users in dependence of each single interest and research. Concerning the quality check for seismic data, as explained in the manuscript, MUDA assures the interoperability with ISMDq (https://ismd.mi.ingv.it), that is the quality DB for the Italian seismic stations.**

5) there is no information about logs of visits to individual wells and equipment maintenance work.

**We are sorry, but these (useful) technical information are in our opinion off target.**

6) The authors of MUDA carried out additional processing of the original seismic records with a frequency of 100 and 200 Hz in order to ensure their loading and viewing in MUDA. In my opinion, this is an unnecessary procedure that does not have a scientific basis for solving problem No. 1. Data from earthquake catalogs would be sufficient to solve it.

**Sorry in advance, but, in this case we don't agree with the comment: it is indeed clear that a simple comparison just on the basis of a seismic catalogue is not absolutely sufficient for our scope and for scope of MUDA. At first, the seismic catalogues are not always complete and often (if not always) they are characterized by a minimum magnitude threshold, that in many cases not include the microseismicity, fundamental in our context. At second, the recorded transients can be natural (e.g. earthquakes or other environmental phenomena) or anthropic and each one it is characterized by a typical frequency content, so that just with a frequency domain analyses it is possible to discriminate the origin of the recorded transients. Moreover, the proposed frequency analyses is not important just for earthquakes detection, but, for example, for other important issues concerning the so named environmental seismicity. As an example, it could be used for landslides monitoring, such as in case of Bondo site (see Ferrari et al., 2024, see figure 12 and related explanation) where a clear correlation was found among seismic transients due to landslides and strong variation of water in well. We think that the proposed event detection analysis in time and in frequency domains is in this case appropriate and necessary.**

7) Similar remarks arise when solving the problem of searching for correlations between the behavior of geochemical parameters - radon and carbon dioxide with seismicity and meteorological parameters. The results of this analysis also showed that there are very few geochemical observation stations and corresponding data. This should also be written about in the text.

**Your observation is absolutely right, but this aspect is well marked both in the text and above all in the tables 1, where it is clear the number and type of considered stations and their related period of recordings.**

8) The presentation of graphical material in the manuscript and in MUDA is mostly unsatisfactory: small, unclear font makes it difficult to see dates; on the Time scale, years are not indicated; there are errors in the signatures (Fig. 4, 5 – "Idrogeochemical???"; the dates of inspection of well bores are not indicated. Graphs are presented rotated 180 degrees. It is generally very difficult to work with a manuscript, since the text, illustrations and captions are located in different places.

**We agree on the difficulty to work with text, illustrations and captions located in different places, but it is important to remember that this is the format that ESSD request for submission; the same considerations are for rotated figures and graphs. Some figures are rotated as requested by the ESSD editorial team in order to maximize dimensions and quality. The time scale (i.e. years) is missing just in figure 3, where in this case the graphs have a simple descriptive and qualitative meaning concerning the capacity of multiparametric data visualization and comparisons. As suggested the signature errors in figures 4 and 5 are corrected.**

You will find all modifications marked in yellow in the revised version of the manuscript that we will submit to ESSD web portal, after collecting all reviewers suggestions.

Best regards, Marco Massa

Dear Dr. Petrosino,

thank you for the attention to our work and for your careful review.

Concerning the minor comments and suggestions, we indicate below the related answers (in bold):

1) The authors generally use the word "Radon" in uppercase, but sometimes it is written in lowercase. Please make it homogeneous.

**Done**

2) L 31-32 Unclear sentence.

**The sentence has been modified, as suggested.**

3) L 39 evidence --> evidences

**Done**

4) L 86 Po alluvial basin and Northern and Central Apennine chains --> Po alluvial basin, Northern and Central Apennine chains

**Done**

5) L100 October 30th, 1901 à October 30, 1901. Check and make homogeneous the date format in the whole text (the authors mainly use Month day, year). See also L583-584 and the captions of Figure 7, 8, 10 and 11.

**Done. The date format is now homogeneous.**

6) L247 Hydrogeochemical, seismic, meteorological and Radon data. I guess the authors should include CO2 too.

**We agree with your comment, but in this particular case it is not correct to modified the text because currently the CO2 measurements are set one to every hour without the chance to perform a resampling. The selected interval for recordings arose from a compromise between the usefulness and reliability of the recorded data, the efficiency of the solar panel recharge (also in case of stormy periods) and the no optimal LTE connection for real time data transmission. Of course, as suggested, in a more better conditions the CO2 measurements could be set with lower intervals of recordings (e.g. 30 min, 15 min), even if with the attention to avoid too short and unreliable measurements.**

7) L351 RSM --> RMS. Check also in the remaining part of the text (e.g. L506)

**Done**

8) L356 Root Mean Square --> RMS

**Done**

9) The abbreviation has been previously defined, so please use it in the remaining part of the text (e.g. L397). This also holds for FFT (Fast Fourier Transform).

**Done**

10) Section 4.2 regards seismic data processing, but part of this procedure has been described at the end of Section 4.1 (Processing of row data). Although the description of seismic data processing is longer compared to the other geochemical and hydrological data, I suggest to merge section 4.2 with the last paragraph of Section 4.1. Otherwise, the authors could make 5 different subsections relative to the processing of each of the 5 types of data.

**As suggested, we have included the seismic data processing in the subparagraph 4.1, of consequence, some parts of the old paragraph 4.2 have been shortened and remodeled.**

11) L425 HVSR, please add a reference to this technique.

**Done. Moreover a new reference has been added in the list.**

12) L 442 A further 2 --> 2 further

**Done**

13) L493 – 499 and L581. 2020-12-29… 202/12/19… Make homogeneous the date format.

**Done. The date format is now homogeneous.**

14) L542 session --> section

**Done**

15) L550-551 as a consequence of 2 days of intense rainfall with measured values of precipitation in a narrow area surrounding Bondo up to 400 mm --> as a consequence of 2 days of intense rainfall with measured values of precipitation up to 400 mm in a narrow area surrounding Bondo

**Done**

16) Section 7 (Data Availability) should me moved (and renumbered) after Section 8 (Usage Notes and conclusions).

**Done**

17) L644 acquifers features --> acquifer features (or acquifers' features)

**Done**

18) L649 In particular --> In addition

**Done**

19) L1135 Multiparametric sites web page à Multiparametric site web page (or Multiparametric sites' web page)

**Done**

20) L1149 type à types

**Done**

21) L 1150 starts à stars

**Done**

You will find all modifications marked in yellow in the revised version of the manuscriot that we will submit to ESSD web portal in a few days, after collecting all reviewers suggestions.

Best regards, Marco Massa

---

## Referee Report (RR1)

Review of the manuscript by Galina Kopylova
Title: MUDA: dynamic geophysical and geochemical MUltiparametric DAtabase
Author(s): Marco Massa, Andrea Luca Rizzo, Davide Scafidi, Elisa Ferrari, Sara Lovati, Lucia Luzi, and the MUDA working group
MS No.: essd-2024-185
MS type: Data description paper
Iteration: Revised submission

Dear Editor!

The authors have taken into account, where possible, the comments made on the quality of the data presented in the database. Useful additions to Table 1 and Table 2, as well as the text (L 464-474, 557-558 and others) has been made. The quality of the illustrations and explanations to them has been significantly improved.

I believe that the edited manuscript can be published in the journal.

With respect,
Galina Kopylova